# Evaluating the Arrhenius equation for developmental processes

Joseph Crapse[1,2,3] (iD), Nishant Pappireddi[2,3], Meera Gupta[2,3,4] (iD), Stanislav Y Shvartsman[1,2,3,5], Eric Wieschaus[1,2,3,*] (iD) & Martin Wühr[1,2,3,**] (iD)

## Abstract

The famous Arrhenius equation is well suited to describing the temperature dependence of chemical reactions but has also been used for complicated biological processes. Here, we evaluate how well the simple Arrhenius equation predicts complex multi-step biological processes, using frog and fruit fly embryogenesis as two canonical models. We find that the Arrhenius equation provides a good approximation for the temperature dependence of embryogenesis, even though individual developmental intervals scale differently with temperature. At low and high temperatures, however, we observed significant departures from idealized Arrhenius Law behavior. When we model multi-step reactions of idealized chemical networks, we are unable to generate comparable deviations from linearity. In contrast, we find the two enzymes GAPDH and β-galactosidase show non-linearity in the Arrhenius plot similar to our observations of embryonic development. Thus, we find that complex embryonic development can be well approximated by the simple Arrhenius equation regardless of non-uniform developmental scaling and propose that the observed departure from this law likely results more from non-idealized individual steps rather than from the complexity of the system.

**Keywords** Arrhenius equation; *Drosophila melanogaster*; embryonic development; temperature dependence; *Xenopus laevis*

**Subject Category** Development

**Mol Syst Biol. (2021) 17: e9895**

## Introduction

For more than a century, the Arrhenius equation has served as a powerful and simple tool to predict the temperature dependence of chemical reaction rates (Arrhenius, 1889). This equation, named after physical chemist Svante Arrhenius, posits that the reaction rate

($k$) is the product of a pre-exponential factor (A) and an exponential term that depends on the activation energy ($E_a$), the gas constant (R), and the absolute temperature (T).

$$k = A e^{\frac{-E}{RT}} \qquad (1)$$

In the late 19th century, scientists proposed many relationships between reaction rates and temperature (Berthelot, 1862; Schwab, 1883; Van't Hoff & Hoff, 1884; Van't Hoff, 1893; Harcourt & Esson, 1895). The Arrhenius equation would come to stand out from the rest, in part because it could be intuitively interpreted based on transition-state theory (Evans & Polanyi, 1935; Eyring, 1935a, 1935b; Laidler & King, 1983). Based on this theory, the exponential term of the Arrhenius equation is proportional to the fraction of molecules with energy greater than the activation energy ($E_a$) needed to overcome the reaction's energetically unfavorable transition state. The pre-exponential "frequency" factor A can be physically interpreted as proportional to the number of molecular collisions with favorable orientations.

The Arrhenius equation's use has also been extended to more complex biological systems, such as frog, beetle, and fly development, occasionally finding non-Arrhenius behavior (Krogh, 1914; Bliss, 1926; Bonnier, 1926; Ludwig, 1928; Powsner, 1935). More recently, the Arrhenius equation has been investigated for use in describing cell cycle duration (Begasse *et al*, 2015; Falahati *et al*, 2021), or, by extension, the $Q_{10}$ rule modeling proliferation dynamics in populations of bacteria (Martinez *et al*, 2013). Modifications to Arrhenius have even been made, positing mass accounts for deviations from a fairly universal Arrhenius fit (Gillooly *et al*, 2002). This broad applicability of the Arrhenius equation to complex biological systems is surprising given that these systems involve a myriad of reactions, presumably each with its own activation energy and thus temperature dependence.

One of the most complicated biological processes the Arrhenius equation has been applied to was the development of a single fertilized egg into the canonical body plan of an embryo (Chong *et al*, 2018). Embryos of most species develop outside the mother and

1 Undergraduate Integrated Science Curriculum, Princeton University, Princeton, NJ, USA
2 Department of Molecular Biology, Princeton University, Princeton, NJ, USA
3 Lewis-Sigler Institute for Integrative Genomics, Princeton University, Princeton, NJ, USA
4 Department of Chemical and Biological Engineering, Princeton University, Princeton, NJ, USA
5 Center for Computational Biology, Flatiron Institute, Simons Foundation, New York, NY, USA
*Corresponding author. Tel: +1 609 258 5383; E-mail: efw@princeton.edu
**Corresponding author. Tel: +1 617 230 7625; E-mail: wuhr@princeton.edu

many have evolved so that they can adapt to wide temperature ranges. Canonically, it has been observed that embryos develop faster with higher temperature (Khokha *et al*, 2002; Kuntz & Eisen, 2014; Sin *et al*, 2019). However, in the liquid phase separated nucleolus of *Drosophila melanogaster* embryos, some proteins decrease their concentration, and presumably their activity, with increasing temperature (Falahati & Wieschaus, 2017). This general behavior is expected in processes that depend on liquid phase transitions as most intramolecular interactions weaken with higher temperature (Ball & Key, 2014). Recently, it has been proposed that the developmental progression of fly embryo development scales uniformly with temperature, exhibiting Arrhenius-like behavior (Kuntz & Eisen, 2014), which would only be consistent with the Arrhenius equation if all activation energies for rate-limiting transition states are identical. In this case, coupled chemical reactions would collapse into a common Arrhenius equation with one master activation energy and integrated frequency factor, which combines each reaction's frequency factors into one. Is it possible that evolution has led to such uniform activation energies in embryos to enable canonical development over a broad temperature range?

To investigate these questions, we monitored the temperature dependence of developmental progression of fly and frog development. We find that the apparent activation energies of different developmental intervals vary significantly, i.e., the time it takes for embryos to develop through different intervals scales differently with temperature, which is contrary to previous findings (Kuntz & Eisen, 2014). Nevertheless, we corroborate previous findings that the Arrhenius equation still provides a good approximation for the temperature dependence of embryonic development. Lastly, we model coupled chemical reactions and investigate the temperature dependence of individual enzyme activities in an attempt to explain this surprising observation.

## Results

### The Arrhenius equation is a good approximation for the temperature dependence of embryonic progression

To investigate experimentally the temperature dependence of complex biological systems, we acquired time-lapse movies of fly (*Drosophila melanogaster*) and frog (*Xenopus laevis*) embryos from shortly after fertilization until the onset of movement in carefully temperature-controlled environments (Appendix Fig S1). Observing these different embryos allows us to assess the generality of findings as the species are separated by ~1 billion years of evolution (Hedges, 2002). Both species' embryos develop as exotherms and are viable over a wide temperature range. For fly embryos, we recorded developmental progression over 19 events. Throughout this paper, we utilize the 12 most reproducibly scored events (Figs 1 A and EV1A–C, Movie EV1, Dataset EV1). Similarly, for frog we recorded 16 events, and the 12 that we could most reproducibly score were used for further analysis (Figs 1B and EV1D–F, Movie EV2, Dataset EV2). Figure 1C shows for each fly embryo score, the mean times since $t = 0$. Time $t = 0$ is defined as the last syncytial cleavage. We observe a clear trend of decreasing developmental time for increasing temperature (Appendix Fig S2A, Dataset EV3). This inverse relationship between developmental time and

temperatures is in agreement with previous studies (Kuntz & Eisen, 2014). At high temperatures (e.g., 33.5°C), fly embryos are only able to develop until gastrulation and die at this stage (during germ band shortening). Similarly, for temperatures at and below ~9.5°C, fly development arrests after gastrulation (after germ band shortening). We find fly embryos are viable in the temperature range from ~14°C to ~30°C to the last developmental event carefully investigated in this paper, "First Breath", when air first enters the trachea. Figure 1 D shows the developmental times in frog embryos since 3$^{rd}$ Cleavage, which we define as $t = 0$ (Appendix Fig S2B, Dataset EV4). Here too, we see an inverse relationship between developmental time and temperature as previously reported (Khokha *et al*, 2002). Frog embryos are able to develop from ~12°C to ~29°C to "Late Neurulation". Compared to the fly data, our frog data appears to be less monotonic between temperatures, likely due to dissimilarities between clutches of eggs from different females. For technical reasons, frog embryos observed in this study under the same temperature tend to share a common mother while observed fly embryos originated from different mothers.

To investigate how well the temperature dependence observed in both flies and frogs can be captured by the Arrhenius equation, we first obtained developmental rates by inverting time intervals between the scored developmental events. We then generated Arrhenius plots by plotting the natural logarithm of these rates against the inverse of relevant absolute temperatures. If a process strictly follows Arrhenius' equation, it appears linear in the Arrhenius plot. Both the frog and fly data exhibit wide core temperature regions that we approximate with a linear fit, between 14.3 and 27°C in flies and 12.2 and 25.7°C in frogs (Fig 2A and B, and Fig EV2). However, for each organism we observe clear deviations from linearity, particularly outside of these temperature ranges (Fig EV2).

We inferred the apparent activation energies from the slope of the Arrhenius plots for all developmental intervals between adjacent scored events. Figure 2A and B show two example developmental intervals for *D. melanogaster* and *X. laevis*. From these plots, it is apparent that the developmental rates for these interval pairs of each organism show different apparent activation energies (example seen in fly $E_{a, D-E} = 56$ kJ/mol, $E_{a, E-F} = 84$ kJ/mol, *P*-value of $6 \times 10^{-8}$, power of 0.98, *F*-test (Lomax, 2007)). Therefore, these developmental rates scale differently with changing temperatures. In this respect, our results differ from the uniform scaling proposed for fly development in a previous study (Kuntz & Eisen, 2014). However, when we reanalyzed the data that the authors kindly provided, we find that apparent activation energies between developmental intervals vary significantly (*P*-value = $1 \times 10^{-3}$) (Fig EV3A–D, Dataset EV5; Data ref: Kuntz & Eisen, 2014). In our data, the apparent activation energies for all developmental intervals between adjacent scores investigated in fly embryos range from ~54 to 89 kJ/mol, where several intervals differ significantly (Fig 2C, Appendix Fig S3 A and C, and S4A). This is similar to the energy released during the hydrolysis of ATP (about 64 kJ/mol) (Wackerhage *et al*, 1998) and to literature values of enzyme activation energies (~20–100 kJ/mol) (Lepock, 2005). We performed the equivalent analysis in frog embryos shown in figure 2D (Appendix Fig S3B and D). Here, we observe significantly different activation energies ranging from ~57 to ~96 kJ/mol with all early cleavage intervals' activation energies being statistically not different ($E_a = 61$–63 kJ/mol, p-values between 0.51 and 1, *F*-test) (Appendix Fig S4D).

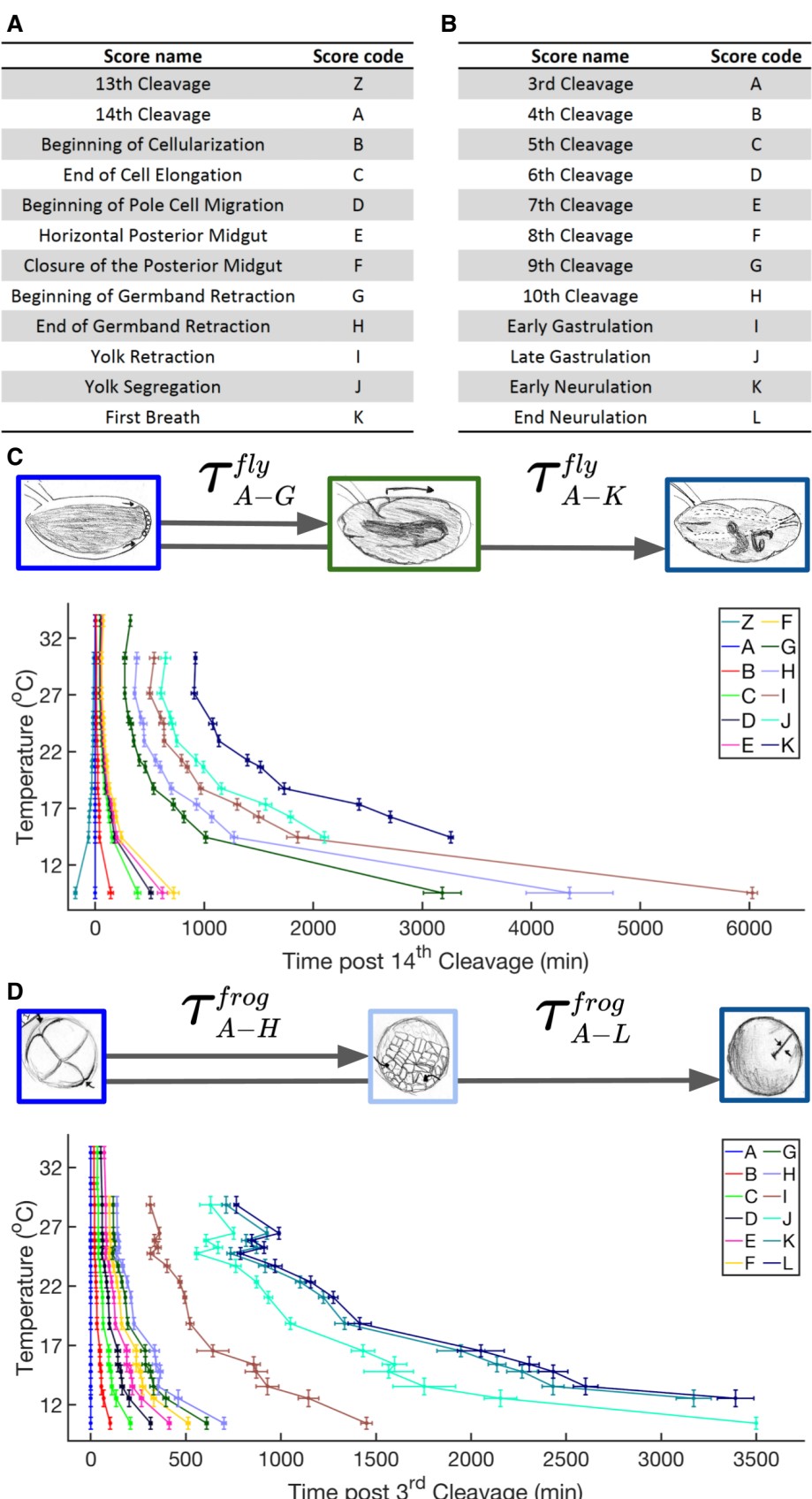

**Figure 1.**

Figure 1.  Temperature dependence of development progression in fly and frog embryos.

A  A table representing *D. melanogaster* developmental scoring event names and sequential score codes used throughout this paper.
B  *X. laevis* developmental scoring and codes used throughout this paper.
C  Shown here is a schematic depicting how time ($\tau$) intervals are determined based on beginning and ending scores. Plotted also are all mean time intervals from $t = 0$, defined as $14^{th}$ syncytial cleavage, to reach various developmental scores in *D. melanogaster* embryos. Error bars in time indicate standard deviation among replicates ($n = 2$–$13$ biological replicates per temperature). Error bars in temperature represent the standard error ($\pm$ 0.5°C) of the thermometer used when recording temperature.
D  As (C) but for *X. laevis*, since $t = 0$ ($3^{rd}$ cleavage) at temperatures ranging from 10.3°C to 33.1°C ($n = 1$–$23$ biological replicates per temperature).

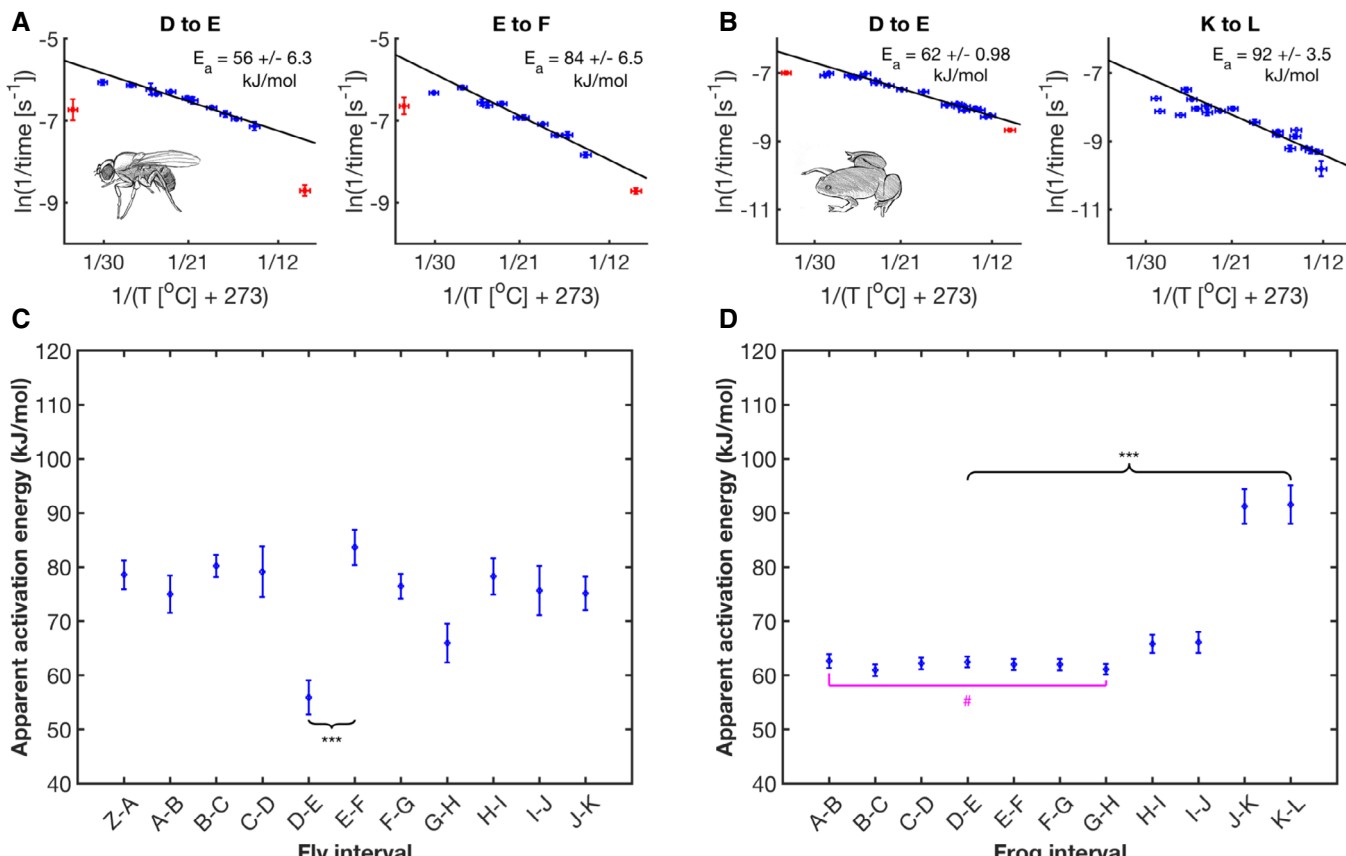

Figure 2.  Apparent activation energies vary significantly between developmental intervals.

A  Arrhenius plots for two examples of developmental intervals (D–E, E–F) in *D. melanogaster*. Blue data points are the means of replicates for viable temperatures that survive until First Breath. Red data represent more extreme temperature values where embryos do not survive until First Breath. A linear regression (solid black line, $n = 66$, $65$ independent biological measurements, respectively) was fit over the core temperature range (14.3–27°C), from which $E_a$ was calculated and reported with its 68% confidence interval. Error bars in temperature represent the standard error ($\pm$ 0.5°C) of the thermometer used when recording temperature. Error bars in ln (rate) represent standard error ($n = 3$–$13$ biological replicates).
B  As (A) but for intervals G-H, K-L in *X. laevis*. Blue data points represent viable temperatures where embryos survive until Late Neurulation. A linear regression (solid black line, $n = 120$, $94$ independent biological measurements, respectively) was fit over core temperatures spanning 12.2–25.7°C, from which the $E_a$ was calculated and reported in black. Error bars in temperature represent the standard error ($\pm$ 0.5°C) of the thermometer used when recording temperature. Error bars in ln(rate) represent standard error ($n = 1$–$10$ biological replicates).
C  Apparent activation energies in fly calculated from Arrhenius plots (Fig EV2A). The *x*-axis is labeled with the developmental interval, marked by start and endpoint. Error bars represent the 68% confidence interval for the activation energy based on linear fit in the Arrhenius plot. Black braces connect examples of developmental intervals that show statistically significant differences in slope (and thus $E_a$), with respectable power ($> 0.8$), ***$P < 0.001$, (*F*-test), ($n = 39$–$60$ independent biological measurements).
D  As (C) but for frog $E_a$ calculated from plots shown in Fig EV2B. Magenta brackets represent groupings (all points above the bracket) showing no statistical difference (#) in activation energy (*F*-test), ($n = 94$–$135$ independent biological measurements).

We expect early cleavage intervals to have equivalent $E_a$'s because of the equivalent processes being performed over each division. Consistency over these intervals therefore lends credence to our analysis and application of the Arrhenius equation. Due to the large evolutionary distance between frogs and flies, we cannot compare equivalent intervals between these two organisms for most

of development. However, the cleavage intervals in frog embryos can naively be expected to be driven by similar biochemical mechanisms as the syncytial cleavages in fly embryos. Interestingly, the corresponding apparent activation energies for cleavage intervals are significantly different (*P*-value = $2.4 \times 10^{-7}$, *F*-test) between the two species with values of ~80 kJ/mol in fly embryos and ~62 kJ/mol in frog embryos. The cause of this divergence is unknown to us but may be due to the differences between syncytial and multi-cellular division mechanisms.

It has previously been proposed by Gillooly et. al that developmental rates should inversely scale with the (embryonic mass)$^{1/4}$. Using the ratio of non-yolk protein content of frog and fly embryos, 25 μg (Gupta *et al*, 2018) and 0.67 μg (Cao *et al*, 2020), respectively, this law predicts frog development to be ~ 2.4-fold slower than fly development. For the few developmental intervals that we can easily map between the evolutionary divergent embryos the observed time-ratios follow the predictions remarkably well, e.g., (at ~22°C): cleavage events (26 min/17 min = 1.5) (this study), onset of gastrulation (540 min/195 min) = 2.8 (Wieschaus & Nüsslein-Volhard, 1986; Nieuwkoop & Faber, 1994), and fertilization to hatching (3,000 min/1,455 min) = 2.1 (Wieschaus & Nüsslein-Volhard, 1986; Nieuwkoop & Faber, 1994; Kuntz & Eisen, 2014). While different developmental intervals show statistically different apparent activation energies this does not interfere with canonical development. Additionally, despite non-uniform scaling for developmental sub-intervals, the rate of embryogenesis from first to last developmental event maintains a Arrhenius-like relationship to temperature.

### Measured departures from Arrhenius law

The apparent activation energies can be inferred from the linear approximation within the core temperature range. Outside of these ranges, the data deviates from idealized behavior. Figure 3A shows two examples for fly embryos: from the 14[th] Cleavage to the Beginning of Germ band Retraction, and First Breath. We confirmed by Bayesian Information Criterion (BIC) analysis that a quadratic fit is indeed more appropriate than a linear fit over the entire temperature range (Wit *et al*, 2012; Dziak *et al*, 2020). We performed identical analyses with all possible developmental intervals between scored events (Fig 3B). We see a clear statistical preference for a quadratic over a linear fit in the Arrhenius plot for all scored developmental intervals in flies. All quadratic fits over the entire temperature range are downward concave (Fig EV2). Reanalysis of the Kuntz and Eisen data reveals similar behavior for developmental intervals marked by their investigated developmental events (Fig EV3E). Figure 3C and D shows the equivalent analyses performed on frog embryos. Here, the data show a similar preference for quadratic over linear fits.

These findings raise the question if the temperature region is also non-linear for the temperature range over which the embryos can develop to the last scored developmental event (14.3–30.1°C in fly and 12.2–28.5°C in frog). We performed BIC analysis for all developmental intervals in fly embryos and find this "viable regime" is clearly quadratic over most intervals (Appendix Fig S4B and C). Although less conclusive, we find similar results when reanalyzing our frog data (Appendix Fig S4E and F). We always observe deviation to be downward concave, i.e., the rates at very

low and very high temperatures are lower than predicted by the Arrhenius equation.

Thus, while the Arrhenius equation is a good approximation for the temperature dependence of early fly and frog development, at temperature extremes, we see clear deviation. This observation supports our initial intuitions that Arrhenius cannot perfectly describe a complex system; although why it deviates and how it is still a fairly decent approximation remains to be answered.

### Multiple steps and nonideal behavior of individual enzymes lead to non-Arrhenius temperature dependence

Next, we investigated what could be the cause for the observed non-linear behavior in the Arrhenius plot for developmental processes. One possibility is that this observed non-linear behavior arises from the complexity of biological systems composed of multiple coupled elementary reactions, each of which follows Arrhenius dependence. To analyze the conditions under which complex networks consisting of many sequential chemical reactions follow the Arrhenius equation, we modeled sequential reactions using a formalism for the relaxation time constant $\tau$ (see Appendix: Mathematical Derivations). For a single reaction, we confirmed the known relationship of $\tau = 1/k$.

However, when we expand the model to a relaxation function with an arbitrary number ($n$) of individual (i) reaction transitions, we find:

$$\tau(T) = \sum_{i=1}^{n} \frac{e^{\left(-E_{a_i}/\mathrm{RT}\right)}}{A_i} \qquad (2)$$

Converting to Arrhenius coordinates ($\tau \rightarrow \ln(k)$ and $1/T$) we arrive at:

$$\ln(k) = -\ln\left(\sum_{i=1}^{n} \frac{e^{\left(E_{a_i}/\mathrm{RT}\right)}}{A_i}\right) \qquad (3)$$

We can clearly see that a sequential multi-reaction series does not yield linear Arrhenius plots. To test whether equation (3) could adequately describe a biological network, we investigated how well we could predict the temperature dependence of a large portion of scored embryonic fly development. Interestingly, when equation (3) is used with actual parameters derived from individual developmental sub-intervals (A–B,…,F–G) the predicted outcome for A–G is still apparently linear, rather than quadratic. This prediction is a near perfect match to the linear fit generated from the experimental data for the same interval A–G (Fig 4A). We observed the same results for frog embryos (Fig EV4).

Even though a system of coupled Arrhenius equations are non-linear, we wondered whether they might appear linear under the temperatures and scenarios applicable to biological phenomena. To this end, we used our equation (3) to simulate 1,000 sequentially coupled chemical reactions; each defined by its own activation energy and prefactor and observed how the reaction rates of the entire network scale with temperature. When we randomly choose 1,000 A's and $E_a$'s among reasonable ranges for biological phenomena (Lepock, 2005), we observe nearly perfect linear behavior in the

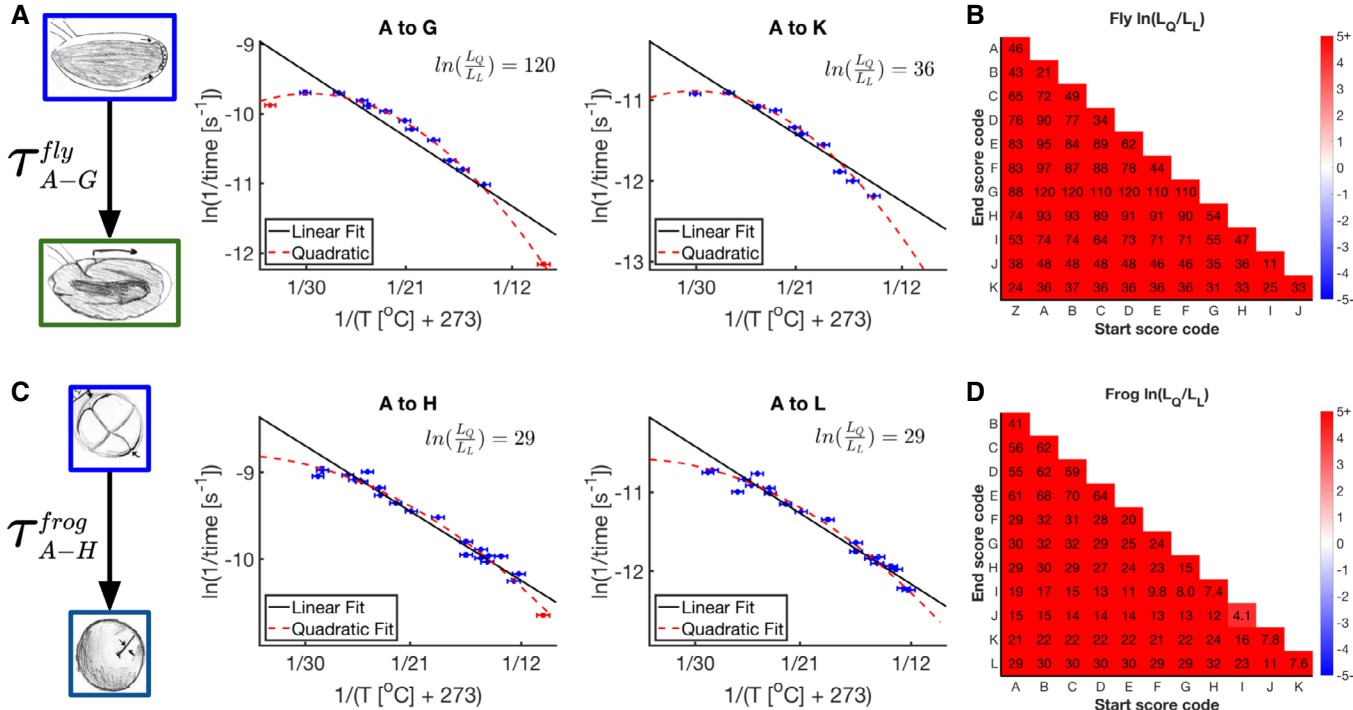

**Figure 3. Fitting the temperature dependence of embryonic development with the Arrhenius equation.**

A  Example Arrhenius plots for fly embryos for the interval from 14th Cleavage (A) to Germband Retraction (G) and First Breath (K), respectively. A linear fit (solid black line) and quadratic fit (dashed red line) were fit over all data points (n = 76 & 43 independent biological measurements, respectively). Error bars in temperature represent the standard error (± 0.5°C) of the thermometer used when recording temperature. Error bars in ln(rate) represent standard error. The log ratio of likelihoods for model selection of quadratic over linear is shown in black.

B  Shown are the natural log ratio of (penalized) likelihoods for quadratic over linear model preference, for all fly developmental intervals marked by their starting event (x-axis) and ending event (y-axis) (n = 42–84 independent biological measurements). Values above 0 indicate that a quadratic fit is preferred to a linear fit.

C  As (A) but for the frog developmental interval from 3rd Cleavage (A) to 10th Cleavage (H) and Late Neurulation (L). Model fits were calculated over all data points (n = 131 and 100 independent biological measurements, respectively).

D  As (B) but for all frog developmental intervals, (n = 97–154 independent biological measurements).

Arrhenius plot for the entire system (Fig 4B). Next, we optimized $E_a$ and A to maximize the curvature of the system, as a proxy for quantifying non-linearity, at T = 295°K using the standard curvature function. This optimization was done while constraining $E_a$ between literature values 20–100 kJ (Lepock, 2005) and constraining the time ($1/k$) for embryonic states between 1 s and 3 days. We chose 1 s for the lower limit as an unreasonably short time in which an embryo could transition through a distinct biochemical state. We used the 3 days of entire fly development in our experiment as an upper time limit for distinct embryonic states. When maximizing curvature for 1,000 coupled reactions using these limits, we observed some non-linearity which we believe could be experimentally detectable (Sawilowsky, 2003; Fig 4B, Appendix Fig S5). Analysis of our equation (3) shows consistent downward concave divergence from linearity for this worst-case scenario (see Appendix: Mathematical Derivations, Concavity). However, even assuming these "worst case" scenarios, our simulations do not show the same level of divergence from linearity observed in our biological data (Fig 4C). Furthermore, in our data we observe a decrease in reaction rates at very high temperatures (Fig 4A). This is impossible to achieve with simulations of coupled reactions that follow the Arrhenius equation, which suggests that factors other than the

coupling of many reactions are likely to contribute to the non-linearity in the Arrhenius plot for developmental processes.

Is it possible that the individual steps, e.g., enzymes, are already non-Arrhenius? To investigate this possibility, we measured the temperature dependence of a reaction catalyzed by GAPDH, a glycolytic enzyme. We chose GAPDH because it is essential to all forms of eukaryotic life and its activity can be easily assayed by following the increase in absorbance of NADH/NAD+ at 340 nm with spectrophotometry. Interestingly, we find that GAPDH shows clearly non-linear behavior in the Arrhenius plot from 10 to 45°C (Fig 4D). When halving the substrate concentrations used, we find similar kinetics, suggesting the enzyme is in the saturated regime (Appendix Fig S6A, Dataset EV6). Additionally, we have assayed another common enzyme, β-galactosidase, monitoring the zero order conversion of ortho-Nitrophenyl-β-galactoside at 420 nm (Fig EV5, Dataset EV7 and Dataset EV8). Also here, we find that the enzyme shows strong non-linearity in the Arrhenius plot. We performed these experiments with saturating substrate concentrations (Appendix Fig S6B). As with our embryonic development data, we find that GAPDH and β-galactosidase activity follows concave downward behavior. Similar to our embryonic data, we find that reaction rates at the high end reduce with increasing

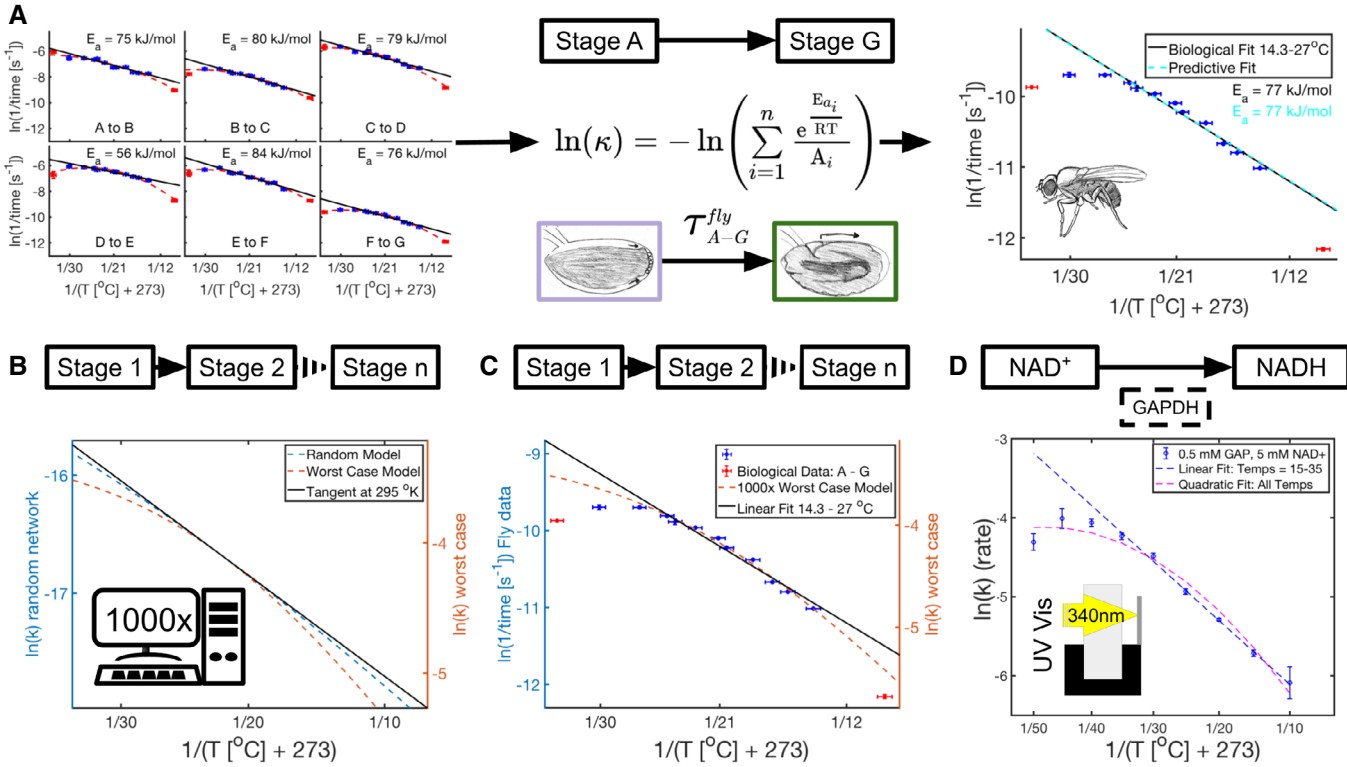

**Figure 4. Complexity and non-idealized behavior of individual enzymes can contribute to non-idealized behavior of developmental processes.**

A  Shown is the methodology used to predict the linear regression for fly development from A to G using empirical parameters from individual intervals and equation (3). Far right, this prediction of ln(k) for the composite network (dashed cyan) is overlaid on the empirical data (blue and red error bars) and linear fit (solid black line) for this developmental interval (A–G). Also shown are the color-coded $E_a$s calculated for each fit over the temperature interval 14.3–27°C ($n = 60$ independent biological measurements). Error bars in temperature represent the standard error ($\pm$ 0.5°C) of the thermometer used when recording temperature. Error bars in ln(rate) represent standard error ($n = 2$–12 biological replicates).

B  Shown is a schematic of a multi-reaction network from Stage 1 to Stage n. Comparison of two reaction networks modeling equation (3) with 1,000 coupled reactions, one with randomly selected $E_a$ and A (dashed blue line), the second with $E_a$ and A optimized for maximum curvature at 295°K (dashed orange line). To allow direct comparisons, the y-axes were scaled to result in overlapping tangents calculated at 295°K (solid black line).

C  As (B), however, the worst-case model (dashed orange) is compared to biological data (blue and red error bars representing standard error in ln(rate), $n = 2$–12 biological replicates per temperature) from (A). To allow direct comparisons, the y-axes were scaled to result in overlapping linear fits over 14.3–27°C (solid black line).

D  Shown is a schematic representing the conversion of NAD+ to NADH via GAPDH catalyzation. GAPDH's conversion of $NAD^+$ to NADH was monitored with UV/VIS spectroscopy at 340 nm. Plotted here is the Arrhenius plot for this conversion at various temperatures between 5°C and 45°C. Means of technical replicates (blue circles) are fit with a linear fit (dashed blue line) from 15 to 35°C and a quadratic fit (dashed magenta) over the entire temperature range. Standard error is shown as blue error bars ($n = 2$–4 technical replicates per temperature).

temperature (Figs 4A and D, and EV5). At the very high end, the enzyme is likely starting to denature (Daniel *et al*, 1996). However, denaturing is unlikely to explain the non-idealized behavior at lower temperatures. It has been proposed that such downward concave behavior in the Arrhenius plot could be due to changes of rate-limiting steps as a function of temperature (Fersht, 1999) or due to lower heat-capacity of the transition state versus enzyme substrate complex (Hobbs *et al*, 2013; Arcus *et al*, 2016; Arcus & Mulholland, 2020). Enzymes have likely evolved to work optimally at a certain temperature (39°C for rabbit GAPDH used in this assay). Therefore, deviating to lower or higher temperatures might lead to individual enzymes' reaction rates being lower than when assuming idealized Arrhenius behavior. Consistent with this, all Arrhenius plots shown for developmental intervals, or this individual enzyme, are statistically significantly quadratic with downward concave shape.

# Discussion

The Arrhenius equation is used for simple chemical reactions to relate the reaction rate with the energy necessary to overcome activation barriers, i.e., activation energy. We find that embryonic fly and frog development can be well approximated by the Arrhenius equation as has been previously proposed by others (Kuntz & Eisen, 2014; Chong *et al*, 2018). By examining the data more carefully, we observe that the relationship between temperature and developmental rates in both species is confidently described by a concave downward quadratic in the Arrhenius plots.

One striking finding of our study is that different developmental processes within the same embryo clearly scale differently with varying temperature, i.e., the apparent activation energies for different developmental intervals can vary significantly. We reaffirmed this observation upon reanalyzing Kuntz & Eisen's, 2014 data

(Fig EV3). Different temperature scaling has also previously been observed in component processes in presumed simpler processes such as cell cycle progression during the cleavage division in fly embryos (Falahati *et al*, 2021).

Our modeling studies demonstrate that, in principle, linking multiple Arrhenius-governed reactions can only lead to concave downward behavior as previously shown by Roe *et al* (1985). However, when we reasonably limit k and $E_a$, coupling of sequential reactions can only modestly contribute toward the observed non-idealized behavior. In contrast, when we observed the temperature dependence of a single enzymatic model process, as in our assays of GAPDH or β-galactosidase, we observed clear non-linear, concave downward behavior. We therefore propose that the observed non-linearity of developmental rates is due to individual non-idealized rate-limiting processes rather than the coupling of multiple developmental processes with different activation energies.

However, several other factors may also contribute to this behavior. Although we have observed non-linearity over temperature ranges where morphology is normal and viability is high, it is possible that additional processes come into play at extreme temperatures. Acute temperature stress responses utilizing specialized mechanisms may modulate the cell and molecular scale events occurring during development. Embryos might activate entirely different pathways at more extreme, near non-viable, temperatures, e.g., via cold or heat stress.

One major question this study raises is how complex embryonic development can result in a canonically developed embryo if the different reactions required for faithful development proceed at different relative speeds at different temperatures. In our assays, we are only able to follow temporally sequential reactions, and one can argue that increasing or decreasing time spent at a particular event should not influence the success of development. However, development must be much more complex and hundreds or thousands of reactions and processes must occur in parallel, e.g., in different cell types developing at the same stage. Therefore, how can frog and fly embryos be viable over a ~15°C temperature range wherein different developmental intervals' varying temperature sensitivity could possibly throw development out of balance? We envision two major possible developmental strategies to overcome this problem. Either all rate-limiting steps occurring in parallel at a given embryonic stage have evolved similar activation energies, or the embryos have developed checkpoints that assure a resynchronization of converging developmental processes over wide temperature ranges.

## Materials and Methods

### *Drosophila melanogaster* data collection and analysis

*Drosophila melanogaster* mutants for klarsicht were maintained as previously described (Wieschaus & Nüsslein-Volhard, 1986; Jäckle & Reinhard, 1998). Flies were allowed to lay eggs for an hour, at which time fresh embryos were collected for time lapses. Embryos were submerged in halocarbon oil 27 (Sigma Cat# H-8773) and selected if they were retracted from the posterior vitelline membrane, signaling successful fertilization.

3–4 selected fly embryos were mounted in halocarbon oil on a slide with an oxygen permeable membrane (Kenneth Technology,

Biofoil #03-670-814). A glass coverslip was gently rolled over the embryos to orient them in lateral view. Slides were placed on transmitted light bright field microscopes set at 20× magnification in temperature controlled rooms for between 9.4 and 33.4°C. Microscopes were focused on a single embryo with the best orientation for imaging. Images were acquired with one of the following cameras: Canon Rebel Ti5/6, Swiftcam 3 Megapixel, OMAX 9.0 MP, or AmScope MU300. Time lapses were recorded from syncytial cleavages until embryo hatching.

To record and validate temperatures for the fly embryo data collections, temperatures were taken next to each a microscope's sample holders (~3 cm from the embryo) using either an Elitech RC-5 (standard error ± 0.5°C), Dickson TH300 (standard error, ± 1.0°C), or Fluke 54 II B (standard error, ± 0.3°C) thermometer. We worked with two microscopes in the room. When comparing the temperatures between microscopes, they never differed more than a degree suggesting the temperature in the room was very homogenous.

For data analysis, time lapse images were converted into video files with absolute time stamped on each frame and manually scored based on the scoring metric depicted in Fig EV1 and described below. Videos are available at the ASCB image library: http://cellimagelibrary.org/groups/53322.

Videos were scored based on video absolute time. Developmental interval timings were then calculated relative to 14th Cleavage (time zero). To adjust these times, the 14th Cleavage absolute time was subtracted from all subsequent (or previous) score times. Descriptions of how each developmental score considered was scored are as follows, in chronological order. The coefficient of variation analysis used to determine which scores were used throughout this paper (lettered scores) and those which were dropped (numbered scores) can be found in Fig EV1.

Z – 13th Cleavage: The end of the 13th syncytial nuclei division. Scored as the point when the surface flow associated with the division reaches the posterior end of the egg and begins to retract.

A – 14th Cleavage: The end of the final syncytial retraction wave that marks the last nuclei division before cellularization. Scored as the point when the flow begins to retract from the posterior end.

B – Beginning of cellularization: Scored as the point when cell membranes begin to form and nuclei become visibly distinct from the outermost edge of the embryo as the membranes move inward.

C – End of Cell Elongation: Scored when the cellularization front completes its inward procession and the ventral cells begin a contracting movement.

D – Beginning of Pole Cell Migration: The pole cells begin their migration to the dorsal side of the embryo. Scored at the first movement toward the dorsal side.

E – Horizontal Posterior Midgut: Scored when the floor of the posterior midgut supporting the pole cells appears horizontal as compared to the anterior–posterior axis of the embryo.

F – Closure of the Posterior Midgut: Scored once the pole cells fulling invaginate into the interior of the embryo and the surrounding tissue appears to pinch shut.

G – Beginning of Germ band Retraction: The germ band on the dorsal side of the embryo begins to retract posteriorly, scored when its posterior end begins to move from its anterior position.

H – End of Germ band Retraction: Germ band is now fully retracted and its posterior end has reached the posterior end of the embryo.

I – Yolk Retraction: The central yolk mass flattens and retracts from the dorsal side of the embryo. Scored when a full separation from the dorsal side is observed.

J – Yolk Segregation: The gut subdivides into three visibly distinct, yolk filled regions.

K – First Breath: Scored the first frame that air floods the trachea, as seen by the darkening of the trachea in brightfield.

We also examined the following seven other events listed below as potential developmental interval markers, but found the scoring variable and have not included them in the analysis.

1 – Beginning of Posterior Midgut Opening: Entry into stage 9 with the formation of the stomodeal plate (Wieschaus & Nüsslein-Volhard, 1986).

2 – Head Invagination: Entry into stage 10, stomodeal invagination (Wieschaus & Nüsslein-Volhard, 1986).

3 – Gut Movement: Scored as the point when the gut begins to twitch.

4 – Head Movement: Scored as when the head of the embryo begins twitching.

5 – First Breath Ends: Scored once the trachea are fully darkened.

6 – Gut Contents Migration: Scored when the remaining gut contents (darker material in brightfield) migrates to the most posterior end of the embryo.

7 – Hatching: Scored when the embryo hatches.

### *Xenopus* embryo data collection and analysis

*Xenopus laevis* egg and testis were collected according to previous protocols with the following modifications and according to IACUC animal handling standards, protocol # 2070 (Wlizla *et al*, 2018). Female *X. laevis* frogs were induced using 500 µl of 1,000 U HCG (Human chorionic gonadotropin) about 16 h before egg collection. Females were gently squeezed and eggs harvested dry on a petri dish. A male frog was euthanized in 0.1% aminobenzoic acid ethyl ester (Tricaine, MS222) (Sigma A-5040)) and then sacrificed by pithing. Testes were collected and stored in a 2-ml Eppendorf tube with 1× MMR (Ubbels *et al*, 1983). Later testes were transferred to an oocyte culture media (1 l of OCM; 1 bag of Leibovitz's L-15 Medium powder (Thermo Fisher Scientific #41300039), 8.3 ml Penn/Strep, 0.67 g BSA) with pH adjusted to 7.7 by NaOH and filtered through a 0.22 µm filter (Mir & Heasman, 2008).

Fertilized and de-jellied eggs were prepared as previously described with minor modifications (Ubbels *et al*, 1983). About one quarter of one testis, collected from a male frog, was crushed with a pestle and mixed into 400 µl of 1× Marc's Modified Ringer's, 0.1 M NaCl, 2.0 mM KCl, 1 mM MgSO$_4$, 2 mM CaCl$_2$, 5 mM HEPES, 0.1 mM EDTA (Ubbels *et al*, 1983) to prepare a source of parental sperm. This aliquot was pipetted over about 200 eggs. Using a sterile pestle, the solution and eggs were mixed and incubated at room temperature for 5 min. Embryos were agitated a second time and incubated for another 5 min. Fertilization was then induced using MiliQ H$_2$O. Briefly, embryos were then de-jellied. 50 ml 2% Cysteine de-jellying solution was prepared and its pH titrated to 7.8 with NaOH. MiliQ H$_2$O in the fertilized embryo dish was then exchanged with the de-jellying solution. Embryos were soaked in this solution for 5 min or until the jelly coats appeared to be separated from the embryos. Embryos were

then washed three times with MiliQ H$_2$O. Washes were then repeated with 0.1× MMR and allowed to rest in the final 0.1× MMR wash.

Fertilized and de-jellied embryos were viewed and selected using bright field microscopy for synchronous embryos entering NF stage 2 (first cleavage event; Nieuwkoop & Faber, 1994). Using a custom made wire-loop pipette, the embryos were gently and quickly segregated. NF stage 2 embryos were then gently transferred via large pipette to a new petri dish, described below, containing temperature equilibrated 0.1× MMR.

A 3D printed cover was created using the OnShape.com online software as an.stl file, which was then subsequently 3D printed. The stl file is available on github: https://github.com/wuhrlab/Arrhe niusAndAnimalDevelopment. A simplified side view can be seen in Appendix Fig S1A. A special embryo cage was then constructed using the above cover, a petri dish with 5 mm mesh fused to the petri dish, and sealing clay around the edges (Appendix Fig S1A).

An eyepiece camera (as above for *Drosophila melanogaster* data acquisition) or Ti5 Rebel Canon camera attached to a bright field microscope was set up to capture a developmental time lapse for a specific temperature setting and placed into a temperature controlled temperature chamber (Fisher Scientific, Cat: 97-990E, Mod: 146E). 75 ml of 0.1× MMR was equilibrated, at temperatures between 10.3 and 33.1°C, in the temperature chamber several hours before embryo incubation. The embryo plate was placed in the temperature chamber on a bright field microscope-camera stage beneath the objective and a LED ring lamp. The plate was filled with 25 ml of 0.1× MMR. NF stage 2 embryos were then transferred to the plate and gently positioned on the mesh. The 3D printed cage was then placed over the embryos. The remaining 50 ml 0.1× MMR was pipetted into the cage through the top ventilation holes. A temperature recorder (Elitech RC-5, standard error ± 0.5°C) was placed near the embryo cage on the microscope stage to record temperature over the experiment's duration. Temperature recording was initiated shortly before the temperature chamber was closed and the time-lapse initiated. A time-lapse was then taken using an eyepiece camera (as used above for flies) or a Ti5 Rebel Canon camera attached to a bright field microscope at 30-second to 1-minute intervals for all time courses, until just before hatching.

To validate the temperature experienced by our frog embryos, we used an aquatic thermometer (QTI, DTU6024C-004-C, tolerance provided by the manufacturer ± 0.1°C) that measured the temperature of the 0.1 MMR the embryos were raised. Additionally, we recorded the temperature of the surrounding air in the aforementioned temperature controlled chamber with an Elitech RC-5 temperature recorder (± 0.5°C). We observed that these readings agreed with each other within the standard errors of the thermometers. Each experiment was performed after allowing the controlled temperature chamber to equilibrate for several hours. For the analysis throughout the paper, we used the measured ambient temperature at the microscope stage, directly adjacent to the frog embryos.

For data analysis, time lapse images were converted into video files with absolute time stamped on each frame and manually scored based on the scoring metric depicted in Fig EV1 and described below. Videos are available at the ASCB image library: http://cellimagelibrary.org/groups/54564.

Videos were scored based on video absolute time. Developmental interval timings were then calculated relative to 3$^{rd}$ Cleavage (time zero). To adjust these times, the 3$^{rd}$ Cleavage absolute time

was subtracted from all subsequent (or previous) score times. Descriptions of how each developmental score considered was scored are as follows, in chronological order.

The coefficient of variation analysis used to determine which scores were used throughout this paper (lettered scores) and those which were dropped (numbered scores) can be found in Fig EV1.

A – 3$^{rd}$ Cleavage: Scored when first sign of cleavage furrow became visible.
B – 4$^{th}$ Cleavage: Scored when the first furrows are visible in the four visible animal cells.
C – 5$^{th}$ Cleavage: Scored when the first furrows of the 5$^{th}$ cleavage are visible.
D – 6$^{th}$ Cleavage: Scored when the first furrows of the 6$^{th}$ cleavage are visible.
E – 7$^{th}$ Cleavage: Scored when the first furrows of the 7$^{th}$ cleavage are visible.
F – 8$^{th}$ Cleavage: Scored when the first furrows of the 8$^{th}$ cleavage are visible.
G – 9$^{th}$ Cleavage: Scored when the first furrows of the 9$^{th}$ cleavage are visible.
H – 10$^{th}$ cleavage: Scored when the first furrows of the 10$^{th}$ cleavage are visible.
I – Early Gastrulation: Scored when the animal pole pigmentation starts to move outwards.
J – Late Gastrulation: Scored once a visible wave over the animal pole closed.
K – Early Neurulation: Scored once the anterior neural folds form.
L – Late Neurulation: Scored once the anterior neural folds close.

We also examined the following four other events listed below as potential developmental interval markers, but found the scoring variable and have not included them in the analysis.

1 – Post-Neurulation: Scored when pigment delineates a triangle at the anterior extent of the fused neural folds.
2 – De-Sphericalization: Scored when the embryo begins to elongate.
3 – Contraction of the Hatching Gland: Scored when the pigmented hatching gland appears to contract.
4 – First Anterior Twitch: Scored at the first anterior twitch.

## Generating Arrhenius plots for embryonic time courses and subsequent analysis

A pseudo-reaction rate was determined by inversion of the interval times shown in in Fig 1. The colors chosen to represent the different intervals were calculated using a matlab script (Holy, 2021). The natural log of this pseudo-reaction rate was then plotted against the inverse of absolute temperature (in Kelvin). A linear regression was taken the interval of temperatures that appear most linear (14.3–27°C). The 68% confidence interval of the regression was then extracted for each developmental interval. Linear regressions from different intervals were compared via ANCOVA (an *F*-test) (Keppel, 1991; Lomax, 2007; Tabachnick & Fidell, 2007; Montgomery, 2017). A quadratic fit was also tested and Bayesian Information Criteria (BIC) (Wit *et al*, 2012; Dziak *et al*, 2020) was calculated to compare fits for preference over both the core temperature range and entire temperature range.

Apparent activation energies were compared, and example intervals of significantly different values were highlighted with a curly brace (Sævik, 2021).

Shortly, identical analysis was conducted on frog videos as on fly videos above. Differing from above score times were calculated in relation to 3$^{rd}$ cleavage (time zero) and over the assumed core temperature range of 12.2–25.7°C.

## Simulations of sequential multi-reaction networks

Using the fmincon function and MultiStart in matlab a global optimization was performed to maximize the curvature of our sequential linear reaction series equation (3) as a proxy for quantifying non-linearity. Optimization was done for 2 reactions with constraints on $E_a$ (20–100 kJ (Lepock, 2005)) and on k (1 s to 3 days). The optimized resulting 2-reaction combination with highest curvature at 295 K was expanded to a 1,000 reaction equivalent by expanding the lower $E_a$ reaction to a 999 reaction equivalent with adjusted activation energy. This was possible because multiple Arrhenius reactions can collapse to a single reaction if they share the same $E_a$. Activation energies were adjusted by subtracting the log of the reaction network size minus 1 all multiplied by RT (the new, optimized $E_a$ was calculated as 66.74 − log(netSize-1)*(R*295.15)).

Optimized values for $E_a$ and A were then substituted into equation (3) to predict rates at similar temperature points as investigated in our time-lapse experiments for the associated reaction network size as seen in Fig 4B and C.

For the random simulated network, rand() was used to choose random $E_a$ and k within reasonable bounds (Lepock, 2005). Reasonable k were determined as the inverse of embryonic states between 1 s and 3 days. Rand() results were then fed into equation (3) to predict the overall network as displayed in Fig 4B.

## GAPDH activity assay

We measured GAPDH activity at various temperatures using methods similar to those previously described (Krebs, 1955; Velick, 1955). GAPDH from rabbit muscle was purchased from Sigma (G2267). The assay buffer contained 15 mM sodium phosphate (adjusted to pH 8.5 with HCl), 30 mM sodium arsenate. 26.5 ml of this buffer was mixed with 1 ml 7.5 mM NAD$^+$ (RPI, N30110-1.0), 1 ml 0.1 M DTT, and 1 ml 0.0024 µM of GAPDH. 2.95 ml of this solution was preincubated for 10 min at the given temperature (controlled with a peltier thermostatted cell holder) and absorption measured at 340 nm in a quartz cuvette in an Agilent Cary 300 spectrophotometer. Samples were continuously mixed with a magnetic stir bar. For controls, after pre-incubation we added 50 µl of 15 mM D/L glyceraldehyde-3-phosphate (Caymen, 17865) and continuously monitored absorption at 340 nm for at least five minutes. Analysis was made from minute 1.5 to 4.5 post-addition of D/L GAP. After five minutes of reaction, an additional 100 µl of 0.15 M NAD$^+$ and 100 µl of 15 mM of GAP was added. Analysis was made from minute 2.5 to 4.5 post-addition of additional NAD$^+$ and GAP.

## β-galactosidase activity assay

We measured β-galactosidase activity at various temperatures using methods similar to those previously described (Sambrook *et al*,

1989). β-galactosidase from *Escherichia coli* was purchased from Sigma (48275-1MG-F). A 2.1 ml solution of 105 mM Sodium Phosphate, 1 mM MgCl₂, 10.7 mM ONPG (Sigma, N1127-5G) was prepared. Additionally, 0.05 ml of 2.2 mM β-Mercaptoethanol was added to the above solution. 2.15 ml of this solution was preincubated for 10 min at the given temperature (controlled with a peltier thermostatted cell holder) and absorption measured at 420 nm in a quartz cuvette in an Agilent Cary 300 spectrophotometer. Samples were continuously mixed with a magnetic stir bar. After pre-incubation, we added 0.05 ml of 11 U/ml β-galactosidase and continuously monitored absorption at 420 nm for at least 5 min.

Controls were performed as above, but with a final concentration of 20 mM ONPG.

## Data availability

Fly developmental time-lapses: Cell Image Library server (http://cellimagelibrary.org/groups/53322).

Frog Developmental Time-lapses: Cell Image Library server (http://cellimagelibrary.org/groups/54564).

Modeling and analysis scripts: (https://github.com/wuhrlab/ArrheniusAndAnimalDevelopment).

Expanded View for this article is available online.

## Acknowledgements
We would like to thank Steven Kuntz and Michael Eisen to share the raw data of their 2014 publication for reanalysis. We would like to thank Trudi Schüpbach, Elizabeth Van Itallie, and members of the Wühr and Wieschaus laboratories for helpful suggestions and discussions. This work was supported by NIH grant R35 GM128813 (MW), R01 GM134204-01 (SS), and T32 GM007388 (NP). We are grateful for HHMI support (EW).

## Author contributions
EW, MW, and JC conceptualized the study. JC, EW, and MW performed the experiments. JC, MG, and NP analyzed the data. NP, SYS, JC, and MG developed the analytical framework. EW, MW, and SYS raised funding and supervised the study. JC, MW, and EW wrote the manuscript, and all authors helped edit the manuscript.

## Conflict of interest
The authors declare that they have no conflict of interest.

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
