## [Review Process File · Molecular Systems Biology]

Evaluating the Arrhenius Equation for Developmental Processes

Joseph Crapse, Nishant Pappireddi, Meera Gupta, Stanislav Shvartsman, Eric Wieschaus, and Martin Wühr

DOI: [10.15252/msb.20209895](https://doi.org/10.15252/msb.20209895)

Corresponding author(s): *Eric Wieschaus (efw@princeton.edu), Martin Wühr (wuhr@princeton.edu)*

Review Timeline:

Submission Date:	29th Jul 20
Editorial Decision:	10th Sep 20
Revision Received:	12th Jan 21
Editorial Decision:	22nd Feb 21
Revision Received:	31st May 21
Editorial Decision:	9th Jul 21
Revision Received:	20th Jul 21
Accepted:	21st Jul 21

Editor: Jingyi Hou

Transaction Report:

Thank you for submitting your work to Molecular Systems Biology. We have now heard back from the three reviewers who agreed to evaluate your manuscript. As you will see from the reports below, the reviewers acknowledge the potential interest of the study. They raise however a series of concerns, which we would ask you to address in a major revision.

Since the reviewers' recommendations are rather clear, there is no need to reiterate all the points listed below. Some of the key issues that would need to be addressed are the following:

- Attention should be given to placing the findings in the context of existing literature.
- Given that the reviewers pointed out that the fly data was quite similar to those in the existing studies, additional analyses and experiments would be required to strengthen this study and to provide novel insights beyond what has been done before. For instance, the referees' concerns with regard to the frog data analysis and in vitro enzymatic assays need to be carefully addressed.

All other issues raised by the reviewers need to be satisfactorily addressed as well. As you may already know, our editorial policy allows in principle a single round of major revision and it is therefore essential to provide responses to the reviewers' comments that are as complete as possible.

On a more editorial level, we would ask you to address the following issues:

Reviewer #1:

This is an interesting study of embryonic development and temperature, specifically looking at the tempo or rate of development as a function of temperature. It has long been observed that many biological processes follow a temperature dependence that can be approximated or predicted by the Arrhenius equation. It has been thought that the underlying chemistry governing these biological processes is responsible for the Arrhenius effect. The authors use videomicroscopy to time the appearance of specific morphological landmarks during *Drosophila* and *Xenopus* embryogenesis. They then measure the timing of these events as embryos are raised at different temperatures. The study is fairly rigorous and quantitative.

Previous meta analysis of developmental time and temperature (Gillooly et al 2002 Nature) has indicated that developmental rates and temperature follow the Arrhenius law across the animal kingdom. More recent and careful studies of *Drosophila* by Eisen in 2014 find an Arrhenius like behavior for fly embryogenesis. Interestingly, this present manuscript finds that the apparent E_a for fly embryogenesis is not constant for all stages, which is contradictory to the Eisen study. They find significant but small variation in E_a between stages that might have been missed or overlooked by the previous study. One thing they observed was that the estimated E_a for fly versus frog embryos were somewhat different. Gillooly, West and colleagues had proposed a generalized model for scaling of developmental rate with total body mass - this rate inversely scaled as $M^{-0.25}$. If one normalizes for the comparative mass difference between a frog and fly embryo using this scaling factor, do the values of E_a estimated in this study become more similar? In any event, this could be done and reported, even if it does not appear normalized by mass. The authors should discuss their findings within the context of that West and colleagues study.

They convincingly show that their data does not exactly fit the Arrhenius model but deviates significantly at temperature extremes making a quadratic fit better. This was most convincing for the fly data. However, in Fig 3BD - the BIC scores are lower for frog than fly by an order of magnitude or more. So the quadratic is not really much of a better fit for the frog. Do the authors have an explanation for the difference? The authors should tone down the generality of the non-linearity of the Arrhenius plotting, and not emphasize it is some universal property of animal development.

They briefly explore a couple of explanations why there is nonlinearity. One is a coupled model that I am not equipped to judge, and so will leave to other reviewers. The other explanation is a rather offhand experiment using purified GAPDH (a key metabolic enzyme) to perform enzymatic reactions at different temperatures in vitro. They find a similar nonlinear trend comparing reaction rate to temperature. This result is pretty weak. They use a fixed concentration of substrates and enzyme, not accounting for enzyme kinetic properties. The reaction will display Michaelis Menten kinetics, meaning at certain substrate concentrations relative to enzyme concentration, the rate will be more highly dependent on K_m , the binding affinity of substrates for enzyme. The experiment needs to be done with saturating substrates so it is zero-order. Then the value of rate, V_{max} , will be proportional to k_{cat} , which is the rate they really want to measure. Otherwise, they intrude into the thermodynamics of substrate binding, which can also be temperature dependent but not in the manner of their interpretation. The authors should either:

1. remove this result and experiment
2. codify that the enzyme activities and substrate concentrations used are zero order across the ENTIRE temperature range
3. repeat these experiments with titration of substrate and determine V_{max} across the temperature range.

Other comments:

Fix the Chong et al citation/reference and other references missing information

Use formal and standard means to present equations 1(E0?) and 2 (E1) on page 6 in body of the text.

Define what i and n are in those equations.

In Figs 3 and 4 - what do the error bars represent? And why are there large error bars for temperature? Was it not a highly controlled and invariant variable?

In Methods it is not stated how temperature was measured in the room. Hopefully not relying on a thermostat. Was there a thermometer placed on the stage? More information is needed on how they measured this important variable and how often over the course of an experiment.

Why did they use a *Drosophila* klarsicht mutant for their experiments? Why not a simple wildtype lab strain? Since the mutant affects microtubule-dependent cargo transport inside cells, are the results a possible artefact of the mutant? And what genotype were the fathers? It was not stated.

Show actual images of the frog and fly embryos at the different stages captured from typical movies. I cannot judge the quality of imaging, which is critical if one wants to unambiguously define the time when a particular stage or event is reached. Charcoal sketches of embryos at different stages in the figures are not really acceptable.

Reviewer #2:

Referee Report for: Evaluating the Simple Arrhenius Equation for the Temperature Dependence of Complex Developmental Processes

In this work, Crapse et al., outline an analysis of the temperature dependence of development, particularly with respect to developmental time. They draw on the Arrhenius equation as a description of the temperature-dependent behaviour. They use both invertebrate and vertebrate systems to explore how developmental time varies at different temperatures.

This work builds on recent work (Kuntz and Eisen, 2014, Chong et al. 2018), which quantitatively characterised the temporal development of *Drosophila* embryos. Here, the authors expand on this by providing a more detailed characterisation of the activation energy of different developmental stages. They then demonstrate that the behaviour, while providing a close fit to an Arrhenius-like process, is in fact more correctly described by a more non-linear approximation. Finally, they include analysis of frog early development and draw interesting parallels in the behaviours between the species.

Overall, this is an interesting piece of work that adds substantially to our knowledge of how temperature affects the timing of developmental processes. However, I have a number of concerns that need addressing.

Major issues

The explanation of the experimental results needs significant improvement. In particular, the authors are claiming evidence for relatively subtle variations in developmental time. Yet, I cannot find, in the main paper or the SI, any mention of number of samples. The legends are also similarly opaque. In the SI it is stated that 3-4 fly embryos were imaged at each time. How many in total for each temperature? What was the variability in measurements between independent experiments at the same temperature? Was a power analysis performed to estimate beforehand the number of embryos required for analysis? As shown by Chong et al. there is significant temporal variability that needs to be carefully accounted for. For example, the measured E_a are likely unreliable without sufficient n (likely > 40 embryos per condition). With the current methodology as presented, I do not have confidence that the results are supported with sufficient statistical rigour.

In the modelling, there is reference to the "curvature of the system". However, as far as I can see, this is not defined. Is it referring to the curving away from linearity? This lack of clarity makes this section very challenging to read, as it's not particularly clear what the authors are trying to conclude.

In the simulations, a range of E_A and A are used to simulate different conditions. But, what about stochasticity? It is likely that even two "identical" embryos would show variability in the timing of developmental events due to noise in processes such as transcription factor binding. How does the inclusion of intrinsic noise affect the predictions? This is important, as the authors claim that it is "impossible" to achieve a fit to their data only with Arrhenius-like behaviour. But, they haven't (as far as I can see) accounted for potential stochastic variation that is likely present.

It would help to find a better way to describe the linear and quadratic fittings. The current use of "Arrhenius space" is non-standard and likely to cause more confusion. Given that MSB is a broad biology journal, the authors need to improve the description of these results.

In the Introduction, the authors clearly lay out the open questions and how their work extends on existing results. However, at times there is some redundancy with existing work that should be clarified. For example, the first paragraph of the Discussion states: "We have shown that embryonic fly and frog development can be well approximated by the Arrhenius equation over each organism's core viable temperature range." But this is already well known (Kuntz and Eisen 2014, Chong 2018) and is not the new result of this manuscript. At a minimum, previous work should be more appropriately cited.

Figure 1D, stage "L" in the frog is highly variable. Is this because of low " n ", or is this a consistently reproducible phenotype? Further discussion would help here. Relatedly, the frog data is much noisier than the *Drosophila*. Is this due to low statistical power or is the variability a feature of the early frog development? Currently, given the lack of important details (see point 1), it is difficult to ascertain.

Figure 2C. Given the large error bars (and the uncertainty about number of experiments), only D-E seems like it really could be described as meaningfully different from the other stages. This suggests that E_A is actually quite constrained during development, with perhaps one or two exceptions. It would be interesting to see further analysis of this, particularly why D-E is different. It would be helpful to add the average time between the events and the variability (as quantified in Chong et al. 2018). The error in developmental timing behaves non-trivially with temperature, and this does not seem to have been carefully accounted for here.

I would suggest using the figures from Figure S4 instead in Figure 3. They are more convincing that

the quadratic behaviour is real as they more precisely deal with the question of outliers.

Minor issues

In the last sentence of the abstract, the conclusion given is too strong. While the modelling and experiments support that individual steps, rather than system complexity, drive the Arrhenius-like behaviour, this is far from definitive. This statement needs to be more suitably expressed to acknowledge uncertainty in the results.

The paper is littered with grammatical errors. In the revision, these should be addressed. For example, the first paragraph of the Introduction has multiple tense confusions, making reading more challenging than it should be.

The legends are overly wordy. For example, in Figure 4D, there is a description of why the experiment was done, which effectively repeats what's in the main text. This is unhelpful, as it makes finding out key information about what is actually plotted more laborious.

The references are littered with errors.

Reviewer #3:

One of the striking features about embryonic development is its robustness against genetic and environmental insults. One such insult, of special relevance to species that develop outside their mother, is that of temperature fluctuations. Specifically, various previous experiments had suggested that the timing of development scales with temperature in a manner consistent with the Arrhenius equation. This scaling suggested that development is dominated by one overarching rate-limiting step.

Crapse et al. set out to test whether it is true that overall development scales following the Arrhenius relation. To make this possible, they measured the timing of developmental features during fruit fly and *Xenopus* embryogenesis. They found that, at first glance, the timing between each developmental stage follows an Arrhenius equation, albeit with different parameters suggesting the existence of distinct rate-limiting steps. However, they also identified deviations from this simple scaling at the extremes of the observed temperature ranges, indicating a breakdown of Arrhenius in some of the fundamental reactions dictating development.

While the data for the fly is quite similar to that obtained by Kuntz and Eisen (PLoS Genetics 10:e1004293, 2014), the data for frogs is novel. Further, the BIC-based analysis to determine whether alternative models to Arrhenius can better explain the data, and the simulations meant to prove how multiple rate-limiting steps conspire to dictate the overall temperature scaling of development are novel. These analyses constitute an exciting opportunity to dig deeper in the nature of the fundamental biochemical reactions dictating animal development.

Major Comments:

1) The bulk of this paper focuses on re-doing the experiments done in the Kuntz and Eisen paper using only *Drosophila melanogaster*. They subdivided the developmental stages quite differently and, with these different divisions, found conflicting results with the Eisen paper. Perhaps the

difference just comes down to the fact that the authors are looking at much smaller time windows, as they suggest. Did the authors attempt to compare the Kuntz and Eisen data more directly? We believe those data are publicly available. Additionally, though the authors claim that they looked at a broader range of temperatures, this doesn't seem to be the case. Kuntz and Eisen measured development times between 15-30 °C (and actually began with a wider range of temperatures at the outset), whereas the authors of this paper look at 12-27 °C.

2) The manuscript makes comparisons between the concavity of the different data sets and of the simulations. It would be great to show those comparisons explicitly in, for example, a plot.

3) We found the argument connected to the in vitro enzymatic assay to be somewhat unconvincing. Though it's true that many enzymes do not have an Arrhenius-like relationship to temperature across their full range of activity, and they specifically show that to be the case for GAPDH, the direct connection to the temperature-dependence of the overall developmental rate seems tenuous. Why do the authors assume that the underlying molecular scale processes are the same across all temperatures, including stress conditions? What if heat stress or cold does something beyond affecting the rates of the processes that occur at the core temperatures by activating different pathways entirely? Additionally, when considering coupled reactions, which step is rate limiting often depends on the temperature because differences in activation energy can cause different steps of a single multi-step process (e.g. transcription) to be slower at different temperatures. This causes the concave downward temperature relationship described, even though it arises from the coupling of two Arrhenius temperature relationships as shown, for example, by Roe et al., *J Mol Biol* 184:441 (1985).

Minor comments:

1) Line numbers would have been helpful.

2) There are various typos and grammatical errors throughout the manuscript.

3) Broken reference: Chong et al.

4) Is the "integrated frequency factor" defined anywhere in the text?

5) Figure 1A: Is it "yolk" instead of "yoke"? "Yoke" is mentioned in the movie as well.

6) Much of the important data are presented in Figures 1C and D, but we found them extremely hard to read, even when they are blown up in one of the supplementary figures. The color scheme and the legend are also amplifying this particular issue. Could the authors plot them with a log scale on the x-axis or as a proportion of total developmental time as was done in the Kuntz and Eisen paper?

7) Figure 2C, caption: You're showing the interval, not the endpoint on the x-axis labels, right?

8) Figure 4C: Some graphical way to show the difference in curvatures to make it clear that there is a modest divergence would be helpful.

9) The methods section needs to be significantly overhauled:

9.1) The math is very hard to follow with variables left undefined in the text before they appear in

the equations.

9.2) The computational approach is not described in sufficient detail.

9.3) Figure S2A: What's PMG?

9.4) Figure S5A: The labels on the plot are pretty obscure.

9.5) The description of temperature control and monitoring throughout the experiments was also insufficient in our opinion. Why did the authors abandon the microfluidic setup used in the same research groups for previous papers? Further, given that the room temperature was controlled, did the authors also monitor the sample temperature? How is the temperature of acclimated water monitored and maintained?

Response to Reviewers:

Reviewer #1:

This is an interesting study of embryonic development and temperature, specifically looking at the tempo or rate of development as a function of temperature. It has long been observed that many biological processes follow a temperature dependence that can be approximated or predicted by the Arrhenius equation. It has been thought that the underlying chemistry governing these biological processes is responsible for the Arrhenius effect. The authors use videomicroscopy to time the appearance of specific morphological landmarks during *Drosophila* and *Xenopus* embryogenesis. They then measure the timing of these events as embryos are raised at different temperatures. The study is fairly rigorous and quantitative.

We thank the reviewer for his/her thoughtful comments. Please find detailed responses to the individual points raised below shown in blue.

Previous meta analysis of developmental time and temperature (Gillooly et al 2002 Nature) has indicated that developmental rates and temperature follow the Arrhenius law across the animal kingdom. More recent and careful studies of *Drosophila* by Eisen in 2014 find an Arrhenius like behavior for fly embryogenesis. Interestingly, this present manuscript finds that the apparent E_a for fly embryogenesis is not constant for all stages, which is contradictory to the Eisen study. They find significant but small variation in E_a between stages that might have been missed or overlooked by the previous study. One thing they observed was that the estimated E_a for fly versus frog embryos were somewhat different. Gillooly, West and colleagues had proposed a generalized model for scaling of developmental rate with total body mass - this rate inversely scaled as $M^{-0.25}$. If one normalizes for the comparative mass difference between a frog and fly embryo using this scaling factor, do the values of E_a estimated in this study become more similar? In any event, this could be done and reported, even if it does not appear normalized by mass. The authors should discuss their findings within the context of that West and colleagues study.

Thank you for the suggestion. Please note that when scaling developmental rates by mass as proposed by Gillooly et al. the scaling only affects the intercept ($\ln A$) in the Arrhenius plot but not the E_a values. Based on the reviewer's suggestion we asked if the developmental timing difference between fly and frog embryos follows the scaling law proposed by Gillooly. Using the ratio of non-yolk protein content of frog and fly embryos (25ug (Cite [bookhttps://doi.org/10.1007/978-1-4939-8784-9_13](https://doi.org/10.1007/978-1-4939-8784-9_13)) and 0.67ug

(DOI:<https://doi.org/10.1016/j.celrep.2020.107783>) the scaling law predicts that fly development would be $(25/0.67)^{0.25} \sim 2.4$ fold faster than frog embryos. Because of the large evolutionary distances only few developmental stages can be mapped between frog and fly embryos. For those stages the the observed time-scaling seems to be well approximated by the proposed scaling (e.g. at $\sim 22^{\circ}\text{C}$): cleavage stages (26min/17min = 1.5) (this study), onset of gastrulation (540min/195min) = 2.8 (ISBN 0-8153-1896-0; ISBN-10: 0947946454), and fertilization to hatching (3000min/1455min) = 2.1 (ISBN 0-8153-1896-0; doi:10.1371/journal.pgen.1004293/, ISBN-10: 0947946454).

We thank the reviewer for pointing out the previous work and have added this interesting observation to the discussion when comparing apparent activation energies between fly and frog embryos, as seen in our main text on line 176:

“It has previously been proposed by Gillooly et. al that developmental rates should inversely scale with the (embryonic mass)^{0.25}. Using the ratio of non-yolk protein content of frog and fly embryos (25ug (PMID: 30151767) and 0.67ug (<https://doi.org/10.1016/j.celrep.2020.107783>) this law predicts frog development to be ~ 2.4 fold slower than fly development. For the few developmental stages that we can easily map between the evolutionary divergent embryos the observed time-ratios follow the predictions remarkably well (e.g. at $\sim 22^{\circ}\text{C}$): cleavage stages (26min/17min = 1.5) (this study), onset of gastrulation (540min/195min) = 2.8 (ISBN 0-8153-1896-0; ISBN-10: 0947946454), and fertilization to hatching (3000min/1455min) = 2.1 (ISBN 0-8153-1896-0; doi:10.1371/journal.pgen.1004293/, ISBN-10: 0947946454).”

They convincingly show that their data does not exactly fit the Arrhenius model but deviates significantly at temperature extremes making a quadratic fit better. This was most convincing for the fly data. However, in Fig 3BD - the BIC scores are lower for frog than fly by an order of magnitude or more. So the quadratic is not really much of a better fit for the frog. Do the authors have an explanation for the difference? The authors should tone down the generality of the non-linearity of the Arrhenius plotting, and not emphasize it is some universal property of animal development.

We believe the BIC values in the previous submission were more convincingly non-linear for fly over frog mostly due to the more consistent data-quality of our fly data compared to the frog data. We observed that frog embryo developmental progression seems to be somewhat clutch dependent (embryos from the same mother tend to develop with similar speeds). For technical reasons, in our study embryos observed under the same temperature often share the same mother frog, while fly embryos observed under the same temperature tend to come from different mothers. We added the following statement to line 130 in our manuscript explaining this:

“Compared to the fly data our frog data appears to be noisier, likely due to similarities within a clutch. For technical reasons, frog embryos observed in this study under the same temperature tend to share a common mother while observed fly embryos were from different mothers”

We strengthened our frog embryo analysis by collecting and analyzing additional embryos, raising our average n for these BIC calculations from 90 to 135 embryos per developmental stage. Furthermore, we removed some clutches from our data at 18.2 C, where the observed time with developmental progression is clearly discontinuous compared to time-series of other embryos at nearby temperatures. We suspect that this clutch was either consisting of sick embryos or that the temperature control might have been off. In the graph below the embryos that were removed from further analysis are highlighted with a red rectangle.

Legend: Different colored lines show the developmental time in frog since 3rd cleavage (T₀) for each stage we investigated. Error bars show the standard

deviation among data points. Error in temperature is the standard error of the temperature recorder as reported by the manufacturer.

Additionally, we noticed that we had poorly scored stage “I” (beginning of gastrulation) initially. For this reason, we rescored stage I for our “old” data. With these improvements the frog BIC values became convincingly more quadratic, as seen in our new figure 3 (line 698):

Figure 3: B) Shown are the natural log ratio of (penalized) likelihoods for quadratic over linear model preference, for all developmental intervals marked by their starting event (x-axis) and ending event (y-axis) (n from 42 – 84). Values above 0 indicate that a quadratic fit is preferred to a linear fit. **D)** As (B) but for all frog developmental intervals (n from 97 – 154).

We also increased our fly embryo sample size from 52 to 73. The additional data strengthened the quadratic BIC preference for fly data. Overall, the fly data remains more convincingly non-linear, as the fly data remains of higher data quality than the frog data. Nevertheless, we believe that we now have strong statistical evidence that both frog and fly data are clearly quadratic as the majority of all developmental intervals are now double digit $\ln(L_Q/L_I)$.

Figure 3: B) Shown are the natural log ratio of (penalized) likelihoods for quadratic over linear model preference, for all developmental intervals marked by their starting event (x-axis) and ending event (y-axis) (n from 42 – 84). Values above 0 indicate that a quadratic fit is preferred to a linear fit.

They briefly explore a couple of explanations why there is nonlinearity. One is a coupled model that I am not equipped to judge, and so will leave to other reviewers. The other explanation is a rather offhand experiment using purified GAPDH (a key metabolic enzyme) to perform enzymatic reactions at different temperatures in vitro. They find a similar nonlinear trend comparing reaction rate to temperature. This result is pretty weak. They use a fixed concentration of substrates and enzyme, not accounting for enzyme kinetic properties. The reaction will display Michaelis Menten kinetics, meaning at certain substrate concentrations relative to enzyme concentration, the rate will be more highly dependent on K_m , the binding affinity of substrates for enzyme. The experiment needs to be done with saturating substrates so it is zero-order. Then the value of rate, V_{max} , will be proportional to k_{cat} , which is the rate they really want to measure. Otherwise, they intrude into the thermodynamics of substrate binding, which can also be temperature dependent but not in the manner of their interpretation. The authors should either:

1. remove this result and experiment
2. codify that the enzyme activities and substrate concentrations used are zero order across the ENTIRE temperature range
3. repeat these experiments with titration of substrate and determine V_{max} across the temperature range.

Thanks for the suggestion. As suggested, we have performed the experiment (3) titrating substrates into the saturated regime and added a similar activity assay for another enzyme (beta Galactosidase). However, we would like to point out that we are agnostic about the molecular mechanism leading to enzyme/embryo kinetic showing nonlinear behaviour in the Arrhenius plot e.g. a change of enzyme substrate affinity with temperature would be entirely consistent with our observations and could contribute to the observed non-linearity in the Arrhenius plot in embryonic development.

We added a supplementary figure showing that over the entire temperature range of our activity assay GAPDH is approximately in the saturated regime. Please note that for technical reasons we were only able to increase the concentration of GAP to ~0.5mM, which is approximately the physiological GAP concentration ((PubMed ID: 2200929, 2200929; 2200929, 4578278). At this concentration, GAPDH seems to be saturated for most over the temperature range assayed, except 50 °C. As shown in the following appendix figure seen on line 67:

Appendix Figure S4: Comparison of enzymatic reaction rates at two different concentrations. A) Shown here are $\ln(k)$ of 0.0028 μM GAPDH at two different substrate concentrations, 0.47 mM GAP and 4.7 mM NAD^+ seen in Fig. 4D vs. 0.24 mM GAP and 2.4 mM NAD^+ . Temperatures at which the assays

were performed are color-coded in the legend. Each data point is representative of 1 sample for each concentration.

To further address the reviewer's comment and to generalize our findings, we additionally added the enzymatic assay of beta-galactosidase activity on Ortho-Nitrophenyl- β -galactoside (ONPG), which we found to be easy to assay in the saturated regime. We ensured that the substrate concentration (10mM) was at saturation throughout the entire temperature range by sampling several temperatures over said range at twice the concentration (and near ONPG solubility limits) (Fig. Appendix S4B, line 71 in the appendix). Also this enzyme (at 0-order) shows strong non-linearity in the enzymatic assay. We would like to stress that these two enzymes are the only ones we tested, both are clearly non-linear in the Arrhenius plot. Shown below are updated graphs for GAPDH (Fig. 4 D) (line 694 in our main manuscript) and beta-Galactosidase (Fig. EV5).

Figure 4: Complexity and non-idealized behavior of individual enzymes can contribute to non-idealized behavior of developmental processes. D) Shown is a schematic representing the conversion of NAD⁺ to NADH via GAPDH catalyzation. Plotted here is the Arrhenius plot for this conversion. Replicates (blue circles) are fit with a linear fit (dashed blue line) from 15 – 35 °C and a quadratic fit (dashed magenta) over the entire viable temperature range. Standard error is shown as blue error bars (n = 2-4). Reaction was run with 0.0028 μ M GAPDH, substrate concentrations are reported in the legend.

We introduce our beta-Galactosidase in our text at line 288 with the following sentences.

“Additionally we have assayed another common enzyme, beta-Galactosidase, monitoring the 0-order conversion of Ortho-Nitrophenyl- β -galactoside at 420 nm (Fig. EV5), where we find similar results to our GAPDH assay. As with developmental data we find that GAPDH and beta-Galactosidase activity follows concave downward behavior.”

Figure EV5: Assay of beta-Galactosidase activity over its viable temperature range under 0-order kinetic conditions. Plotted here is the Arrhenius plot for this conversion. Replicates (blue circles) are fit with a linear fit (dashed blue line) from 15 – 35 °C and a quadratic fit (dashed magenta) over the entire viable temperature range. Standard error is shown as blue error bars (n = 3). Reaction was run with 0.25 U/mL beta-Galactosidase, substrate concentrations are reported in the legend.

Appendix Figure S4: Comparison of enzymatic reaction rates at two different concentrations. B) As (A) but for 0.25 U/ ml B- galactosidase assay seen in Fig. EV 5A for 20 mM vs. 10 mM ONPG.

Other comments:

Fix the Chong et al citation/reference and other references missing information

Thank you for bringing this to our attention - unfortunately the zotero links were broken before we submitted. We went through the paper and made sure that all citations were properly linked and formatted.

Use formal and standard means to present equations 1(E0?) and 2 (E1) on page 6 in the body of the text.

We have gone through the equations and properly represent them according to conventions, labeling the far right edge of the page with (“the equation number”). An example is shown below, update seen at line 241.

$$\tau(T) = \sum_{i=1}^n \frac{e^{(-E_{a_i}/RT)}}{A_i} \quad (2)$$

Define what i and n are in those equations.

We apologize for this omission. We have updated the text to clearly define n and i as the variables used to express the n th equation and “ i ” the max number of segments in the summation. To directly quote the new text seen on line 239:

“However, when the model is expanded to a relaxation function modeling an arbitrary number (n) of individual (i) reaction transitions, we find:”

In Figs 3 and 4 - what do the error bars represent? And why are there large error bars for temperature? Was it not a highly controlled and invariant variable?

Thanks for pointing this out. We added a comment to the figure legend stating that the error bars indicate standard error based on the manufacturer of the used thermometer. We updated the figure legend, seen at line 691 and 708, as follows:

“Error bars in temperature represent the standard error (± 0.5 °C) of the thermometer provided by the manufacturer. Error bars in $\ln(\text{rate})$ represent standard error.”

In Methods it is not stated how temperature was measured in the room. Hopefully not relying on a thermostat. Was there a thermometer placed on the stage? More information is needed on how they measured this important variable and how often over the course of an experiment.

As the reviewer suspects, we controlled the temperature to the best of our ability and monitored temperature stability by placing a thermometer directly next to the sample on the stage. We added additional text to the supplement to clarify this point. As seen starting on line 371:

“A temperature recorder (Elitech RC-5) was placed near the embryo on the microscope stage to record temperature over the experiment’s duration for frog and fly time-lapse collections.”

Why did they use a *Drosophila klarsicht* mutant for their experiments? Why not a simple wildtype lab strain? Since the mutant affects microtubule-dependent cargo transport inside cells, are the results a possible artefact of the mutant? And what genotype were the fathers? It was not stated.

Both fathers and mothers were *klarsicht* mutants. This mutant was used because it is much easier to score developmental stages compared to wildtype embryos, allowing for increased specificity, repeatability, and accuracy. Furthermore, various previous studies suggest

that developmental progression is similar to wild type flies despite divergent biology associated with the mutant (ISBN-10: 0947946454).

Show actual images of the frog and fly embryos at the different stages captured from typical movies. I cannot judge the quality of imaging, which is critical if one wants to unambiguously define the time when a particular stage or event is reached. Charcoal sketches of embryos at different stages in the figures are not really acceptable.

We believe the exaggerated sketches make it easy to understand the scoring criteria in the figures. Of course, this does not replace the need to show the actual data. We have uploaded all time-laps raw data underlying this study to the ASCB imaging server (Frog: <http://cellimagelibrary.org/groups/53201>, Fly: <http://cellimagelibrary.org/groups/53226>). We are still working with the server administrators to improve available image quality and adjust the metadata. In the meantime our data can be accessed via the following anonymous google drive link:

<https://drive.google.com/drive/folders/1zdMgagG6YJo8i7y2haCfq4lan2QSTTP3?usp=sharing>. Please note the drive preview does not display the video in full resolutions. To access high quality movies please download the videos.

Lastly, we provide one example time-lapse for frog and fly development together with this manuscript as supplemental movies (Appendix Movies S1 and S2). In these supplemental movies, we added stamps and halted the movies clearly defining the timing and criteria of the scored stages.

Reviewer #2:

Referee Report for: Evaluating the Simple Arrhenius Equation for the Temperature Dependence of Complex Developmental Processes

In this work, Crapse et al., outline an analysis of the temperature dependence of development, particularly with respect to developmental time. They draw on the Arrhenius equation as a description of the temperature-dependent behaviour. They use both invertebrate and vertebrate systems to explore how developmental time varies at different temperatures.

This work builds on recent work (Kuntz and Eisen, 2014, Chong et al. 2018), which quantitatively characterised the temporal development of *Drosophila* embryos. Here, the authors expand on this by providing a more detailed characterisation of the activation energy of different developmental stages. They then demonstrate that the behaviour, while providing a close fit to an Arrhenius-like process, is in fact more correctly described by a more non-linear approximation. Finally, they include analysis of frog early development and draw interesting parallels in the behaviours between the species.

Overall, this is an interesting piece of work that adds substantially to our knowledge of how temperature affects the timing of developmental processes. However, I have a number of concerns that need addressing.

Major issues

The explanation of the experimental results needs significant improvement. In particular, the authors are claiming evidence for relatively subtle variations in developmental time. Yet, I cannot find, in the main paper or the SI, any mention of the number of samples. The legends are also similarly opaque. In the SI it is stated that 3-4 fly embryos were imaged at each time. How many in total for each temperature? What was the variability in measurements between independent experiments at the same temperature? Was a power analysis performed to estimate beforehand the number of embryos required for analysis? As shown by Chong et al. there is significant temporal variability that needs to be carefully accounted for. For example, the measured E_a are likely unreliable without sufficient n (likely > 40 embryos per condition). With the current methodology as presented, I do not have confidence that the results are supported with sufficient statistical rigour.

Thank you very much for bringing this to our attention. We have made sure that all results are clearly accompanied by their sample number (along with other important statistics and information) and uploaded Appendix Tables 3 and 4 with the time-intervals of the scored embryos.

Additionally we have shown below the power with respect to the number of samples for the given effect size where we claim a significant difference between activation energies (for an

example stage in fly comparing stages D-E to J-K). As seen in the graph a sample size of 10 should be sufficient to give a power well above 0.8, whereas our data for this comparison has a sample size of [100, 135] with an average of 5 embryos per measured temperature spanning ~12-26 °C. We have reported power in the main text where we claim significant differences between stages in flies (line 153).

“example seen in fly $E_{a,D-E} = 56$ kJ/mol, $E_{a,E-F} = 84$ kJ/mol, p-value of 6×10^{-8} , power of 0.98, F test”

Legend: Left) An example showing the relationship between power and p-value to sample size in our fly data. This example is computed using ancova calculations between intervals D-E and E-F over a variable sample size. Right) As left, but for an example in frog comparing intervals D-E and J-K.

Since Chong et. al was investigating the robustness of development from one temperature to another, they performed a power analysis between temperatures, resulting in the necessarily high sample size per temperature condition they estimated.

However, for our research we are comparing the effects of temperature on rates of development for different development stages. For this reason, we believe that the compared conditions are development stages and not temperatures. We have a sufficiently large sample size per developmental stage to give us enough power to claim statistical differences between the stages, for a given effect size. These sample numbers have now been reported in the figure legends. For frogs we have 122 average embryos per condition and for fly we have an average of 60 embryos per condition when calculating E_a .

In the modelling, there is reference to the "curvature of the system". However, as far as I can see, this is not defined. Is it referring to the curving away from linearity? This lack of clarity makes this section very challenging to read, as it's not particularly clear what the authors are trying to conclude.

This is a very good point. We have revised the text to more clearly specify that "Curvature of the system" means a departure from linearity as well as how we defined and quantified such curvature. A quotation from the revised text can be found following beginning on line 262:

"Next, we optimized E_a and A to maximize the curvature of the system, as a proxy for quantifying non-linearity, at $T = 295$ K using the standard curvature function, while constraining E_a between literature values 20-100 kJ (Lepock, 2005) and constraining the time ($1/k$) for embryonic states between 1 second and 3 days."

In the simulations, a range of E_a and A are used to simulate different conditions. But, what about stochasticity? It is likely that even two "identical" embryos would show variability in the timing of developmental events due to noise in processes such as transcription factor binding. How does the inclusion of intrinsic noise affect the predictions? This is important, as the authors claim that it is "impossible" to achieve a fit to their data only with Arrhenius-like behaviour. But, they haven't (as far as I can see) accounted for potential stochastic variation that is likely present.

We have indeed not taken stochasticity into account when previously running our simulations of our model.

To simulate the effect of intrinsic noise on our model we have introduced gaussian noise to the prefactors with a standard deviation of 10%. The 10% is a bit more than the CVs we observe for the variability of time-intervals for our biological data (Reference Figure here). Shown below are the mean and standard deviation (blue error bars) for these simulations. In magenta, we show the underlying deterministic model:

It appears that the simulation of intrinsic noise does not result in a systematic deviation but rather introduces noise around the deterministic model. .

It would help to find a better way to describe the linear and quadratic fittings. The current use of "Arrhenius space" is non-standard and likely to cause more confusion. Given that MSB is a broad biology journal, the authors need to improve the description of these results.

Thank you for pointing this out. We have revised the text removing the mentioning of "Arrheniu Space" in the paper. We have revised the text so as to discuss linear or quadratic fits in our Arrhenius plots such as below (line 317):

"we observe that the relationship between temperature and developmental rates in both species is confidently better described by a concave downward quadratic in the Arrhenius plots, especially when considering the entire temperature range over which the embryos are viable"

In the Introduction, the authors clearly lay out the open questions and how their work extends on existing results. However, at times there is some redundancy with existing work that should be clarified. For example, the first paragraph of the Discussion states: "We have shown that embryonic fly and frog development can be well approximated by the Arrhenius equation over

each organism's core viable temperature range." But this is already well known (Kuntz and Eisen 2014, Chong 2018) and is not the new result of this manuscript. At a minimum, previous work should be more appropriately cited.

We have tried our best to better put our manuscript in the context of the previous literature. For example, we have modified the text above to (line 311):

"We find that embryonic fly and now frog development can be well approximated by the Arrhenius equation over each organism's core viable temperature range as has been previously proposed by others (doi:10.1371/journal.pgen.1004293, <http://dx.doi.org/10.1098/rsif.2018.0304>)."

Figure 1D, stage "L" in the frog is highly variable. Is this because of low "n", or is this a consistently reproducible phenotype? Further discussion would help here. Relatedly, the frog data is much noisier than the Drosophila. Is this due to low statistical power or is the variability a feature of the early frog development? Currently, given the lack of important details (see point 1), it is difficult to ascertain.

Stage L is the latest point in development we score in frog. For this reason developmental issues have more time to accumulate, potentially terminating some of our scored embryos. For this reason it is true that stage L has relatively fewer "n" than other stages, however we do not feel this "n" is low by any standards.

We have additionally taken care to add more data, specifically 27 additional embryos at stage L. Furthermore, we have updated the error bars to the commonly used standard deviations, rather than the previously used 95% (~2x standard deviation). Finally to more adequately represent our data's trend and we combine previous data of similar temperature, correcting the error in time and temperature to account for this combination.

This updated data can be seen in the following:

Figure 1: C) Shown here is a schematic depicting how time intervals for stages are determined based on beginning and ending score. Plotted also are all mean time intervals measured at various temperatures ($n = 2$ to 13 per temperature) since $t=0$, 14th syncytial cleavage, in *D. melanogaster* embryos to reach various developmental scores shown in (A). Error bars in time indicate standard deviation among replicates. Error bars in temperature are the standard error given by the manufacturer of 0.5 °C **D)** As (C) but for *X. laevis*, since T0 (3rd cleavage) at temperatures ranging from 10.3 °C to 33.1 °C ($n = 1$ to 23 per temperature).

Figure 2C. Given the large error bars (and the uncertainty about the number of experiments), only D-E seems like it really could be described as meaningfully different from the other stages. This suggests that EA is actually quite constrained during development, with perhaps one or two exceptions. It would be interesting to see further analysis of this, particularly why D-E is different. It would be helpful to add the average time between the events and the variability (as quantified in Chong et al. 2018). The error in developmental timing behaves non-trivially with temperature, and this does not seem to have been carefully accounted for here.

We apologize for not having clearly defined the error bars in our previous submission. Those indicated 2x standard deviation (~95%). In the updated manuscript we clearly defined each error bar. In figure 2C&D we now use 68% confidence intervals. Together with the additional data acquired the error bars are much smaller than in the previous submission, as seen below with these updated figures.

Figure 2: C) Apparent activation energies in Fly calculated from Arrhenius plots (Fig EV3C). The x-axis is labeled with the developmental interval, marked by start and endpoint. Error bars represent the 68% confidence interval for the activation energy based on linear fit in the Arrhenius plot. Black braces connect examples of developmental intervals that show statistically significant differences, with respectable power (>0.8), in slope (and thus E_a). *** $p < 0.001$, ** $p < 0.01$, * $p < 0.05$ (F-test). **D)** As (C) but for frog E_a calculated from plots shown in Fig EV3D. Magenta brackets represent groupings (all points above the bracket) showing no statistical difference (#) in activation energy (F-test).

To better account for differences in temporal variability at extremes we have taken an additional approach to presenting our data. Performing 5000 bootstrapped regressions we have compiled distributions which better represent the confidence in our reported activation energies. The plots below are now in the Appendix, line 28, as Appendix Fig. S1A, B. Additionally the medians and standard deviation (68% CI) are marked in red and black respectively. The following figures have been scaled consistently between the two to give viewers the ability to compare variation between the two data sets:

Appendix Figure S1: Distribution of bootstrapped apparent activation energies for various developmental periods in fly and frog. A) Histograms of apparent activation energies in Fly calculated from 5000x bootstrapped fits on data used to generate Fig. 2C. Displayed also is the median (dashed red line) as well as 68% confidence (dashed black line). **C)** As (A) but bootstrapping was performed on Frog data used to generate Fig. 2D.

Finally we appreciated the reviewers suggestion to scale figures 2C & D with developmental time between events. Therefore we have rescaled figures 2C & D according to a temperature near each organism middling tolerable range. The updated figure Appendix S1B, D are as follows, seen in the appendix on line 31:

Appendix Figure S1: Distribution of bootstrapped apparent activation energies for various developmental periods in fly and frog. B) As Fig. 2C, but the x-axis has been scaled by mean developmental timings of each stage at

21.1 °C. **D)** As Fig. 2D, but the x-axis has been scaled by mean developmental timings of each stage at 22.2 °C.

I would suggest using the figures from Figure S4 instead in Figure 3. They are more convincing that the quadratic behaviour is real as they more precisely deal with the question of outliers.

We believe that consistent behavior between the two different species observed, as well as corroboration from our enzymatic assays shows that extreme temperatures are indeed a biological phenomenon, rather than statistical outliers. For this reason we feel it important to show non-linearity over the entire organism's viable temperature range that we sampled, rather than arbitrarily choosing a more linear part of temperature range and for this subset testing for non-linearity.

Minor issues

In the last sentence of the abstract, the conclusion given is too strong. While the modelling and experiments support that individual steps, rather than system complexity, drive the Arrhenius-like behaviour, this is far from definitive. This statement needs to be more suitably expressed to acknowledge uncertainty in the results.

We have updated the last sentence of the abstract as follows to address the reviewer's concern:

Original: "Thus, we find that complex embryonic development can be well approximated by the simple Arrhenius Law and propose that the observed departure from this law results primarily from non-idealized individual steps rather than the complexity of the system."

Now as seen on line 28:

Updated: "Thus, we find that complex embryonic development can be well approximated by the simple Arrhenius Equation regardless of non-uniform developmental scaling, and propose that the observed departure from this law likely results more from non-idealized individual steps rather than from the complexity of the system."

The paper is littered with grammatical errors. In the revision, these should be addressed. For example, the first paragraph of the Introduction has multiple tense confusions, making reading more challenging than it should be.

We apologize for the grammatical errors and have done our best to ensure the paper is consistent in regards to tense as well as cleaning up other grammatical issues.

The legends are overly wordy. For example, in Figure 4D, there is a description of why the experiment was done, which effectively repeats what's in the main text. This is unhelpful, as it makes finding out key information about what is actually plotted more laborious.

We have gone through the figures after responding to all the reviewers comments and cut out the redundant material. Hopefully you will find that now we only write information immediately necessary for understanding the figure. Upon cleaning up the figures we also found that critical data had been missing. We have ensured the legends are concise and contain all necessary material to judge the figures compellingness and legitimacy (n, test performed, p values, etc.). For example we have modified figure 4D by deleting all the redundant materials and adding necessary material, with changes as follows:

“To investigate whether non-linear behavior of individual enzymes could contribute to non-idealized behavior of complex biological systems, we investigated the temperature dependence of the activity of GAPDH. GAPDH's conversion of NADH to NAD⁺ can be conveniently monitored with UV/VIS spectroscopy at 340nm. Interestingly, this enzyme shows clearly non-idealized behavior from 9 to 34 °C (vertical gray lines) and beyond, comparable to what we see for embryogenesis as a whole. At very high temperature the rates are actually decreasing with increased temperatures likely due to protein denaturation (orange).”

To simply (line 710):

“Shown is a schematic representing the conversion of NAD⁺ to NADH via GAPDH catalyzed. GAPDH's conversion of NAD⁺ to NADH was monitored with UV/VIS spectroscopy at 340nm. Plotted here is the Arrhenius plot for this conversion at various temperatures between 5 °C and 45 °C, at the following concentrations: 4.7 mM NAD⁺, ~0.47 mM GAP, 0.0028 uM/mL GAPDH (assuming Rabbit GAPDH has a MW of 37kDa [PMID: 10407139]). Replicates

(blue circles) are fit with a linear fit (dashed blue line) from 15 – 35 °C and a quadratic fit (dashed magenta) over the entire viable temperature range. Standard error is shown as blue error bars (n = 2-4).”

The references are littered with errors.

Thank you for pointing this out. Unfortunately, only noticed after our submission that the zotero links in the manuscript were broken. We have fixed these problems.

Reviewer #3:

One of the striking features about embryonic development is its robustness against genetic and environmental insults. One such insult, of special relevance to species that develop outside their mother, is that of temperature fluctuations. Specifically, various previous experiments had suggested that the timing of development scales with temperature in a manner consistent with the Arrhenius equation. This scaling suggested that development is dominated by one overarching rate-limiting step.

Crapse et al. set out to test whether it is true that overall development scales following the Arrhenius relation. To make this possible, they measured the timing of developmental features during fruit fly and *Xenopus* embryogenesis. They found that, at first glance, the timing between each developmental stage follows an Arrhenius equation, albeit with different parameters suggesting the existence of distinct rate-limiting steps. However, they also identified deviations from this simple scaling at the extremes of the observed temperature ranges, indicating a breakdown of Arrhenius in some of the fundamental reactions dictating development.

While the data for the fly is quite similar to that obtained by Kuntz and Eisen (PLoS Genetics 10:e1004293, 2014), the data for frogs is novel. Further, the BIC-based analysis to determine whether alternative models to Arrhenius can better explain the data, and the simulations meant to prove how multiple rate-limiting steps conspire to dictate the overall temperature scaling of development are novel. These analyses constitute an exciting opportunity to dig deeper in the nature of the fundamental biochemical reactions dictating animal development.

Major Comments:

1) The bulk of this paper focuses on re-doing the experiments done in the Kuntz and Eisen paper using only *Drosophila melanogaster*. They subdivided the developmental stages quite differently and, with these different divisions, found conflicting results with the Eisen paper. Perhaps the difference just comes down to the fact that the authors are looking at much smaller time windows, as they suggest. Did the authors attempt to compare the Kuntz and Eisen data more directly? We believe those data are publicly available. Additionally, though the authors claim that they looked at a broader range of temperatures, this doesn't seem to be the case. Kuntz and Eisen measured development times between 15-30 °C (and actually began with a wider range of temperatures at the outset), whereas the authors of this paper look at 12-27 °C.

Despite extensive searching, unfortunately we were unable to locate the bulk of the Kuntz and Eisen datasets in spreadsheet or video form. Because of this, we were therefore unable to make direct comparisons because of the differing scoring criteria used between our paper and theirs.

In regards to the temperature range we apologize for the confusion. We collected data for fly embryos from 10 - 33.1C, which covers a wider range than Kuntz and Eisen. However we analyzed this data in two manners, one which used this temperature range in its entirety to determine the linearity of the data in the Arrhenius plot, and the other which only made use of a subset (12-27 °C as referenced by the reviewer) to determine the apparent activation energies of each stage. To avoid confusion, we updated the manuscript as follows:

Original: "In this respect our results differ from the uniform scaling observed for fly development in a previous study (Kuntz and Eisen, 2014), but this difference most likely reflect the greater morphological resolution possible with our imaging setup and the wider temperature range investigated in our experiments."

Updated (line 155): "In this respect our results differ from the uniform scaling observed for fly development in a previous study (Kuntz and Eisen, 2014), but this difference most likely reflect the greater morphological resolution possible with our imaging setup in our experiments"

2) The manuscript makes comparisons between the concavity of the different data sets and of the simulations. It would be great to show those comparisons explicitly in, for example, a plot.

Thanks for the suggestion. We agree that showing such a direct comparison makes it easier to follow the paper's argument. For this reason we have revised Figure 4, combining B & C (further details below) and then plotting a comparison between the biological data presented in 4A and the simulated worst case scenario in a similar manner as the aforementioned combination.

The new figure 4C is shown below, found at line 715:

Figure 4: Complexity and non-idealized behavior of individual enzymes can contribute to non-idealized behavior of developmental processes. C) As (B) however, the worst-case model (dashed orange) is compared to biological data (blue and red error bars) from (A). To allow direct comparisons the y-axes were scaled to result in overlapping linear fits over 14.3-27 °C (solid black line).

3) We found the argument connected to the in vitro enzymatic assay to be somewhat unconvincing. Though it's true that many enzymes do not have an Arrhenius-like relationship to

temperature across their full range of activity, and they specifically show that to be the case for GAPDH, the direct connection to the temperature-dependence of the overall developmental rate seems tenuous. Why do the authors assume that the underlying molecular scale processes are the same across all temperatures, including stress conditions? What if heat stress or cold does something beyond affecting the rates of the processes that occur at the core temperatures by activating different pathways entirely? Additionally, when considering coupled reactions, which step is rate limiting often depends on the temperature because differences in activation energy can cause different steps of a single multi-step process (e.g. transcription) to be slower at different temperatures. This causes the concave downward temperature relationship described, even though it arises from the coupling of two Arrhenius temperature relationships as shown, for example, by Roe et al., J Mol Biol 184:441 (1985).

Thank you for pointing this out. We have tried to clarify that we do not claim that the observed nonlinearity of temperature dependence for embryonic development has to come solely from the nonlinearity of individual enzymes. We modified the text accordingly and mentioned the alternative possibilities highlighted by the reviewer. This addition can be seen on line 328:

“However, there remains several additional possibilities contributing to this highly complex behavior. For example embryos might activate entirely different pathways at certain temperatures e.g. via cold or heat stress. Additionally, single multi-step processes (e.g. transcription) might exhibit different rate-limiting steps at different temperatures due to the different underlying activation energies ([https://doi.org/10.1016/0022-2836\(85\)90293-1](https://doi.org/10.1016/0022-2836(85)90293-1)).”

Minor comments:

1) Line numbers would have been helpful.

We have added line numbers to the updated manuscript.

2) There are various typos and grammatical errors throughout the manuscript.

Thanks for pointing this out. We have done our best to eliminate typos and grammatical errors.

3) Broken reference: Chong et al.

Our apologies, our reference link broke before submission without us noticing. We have fixed this for the resubmission.

4) Is the "integrated frequency factor" defined anywhere in the text?

Thank you for pointing this out. We never explicitly define what the integrated frequency factor is. We have referenced it as an "integrated" frequency factor (A) based on the number of duplicate reactions present in a collapsed reaction network. We have revised our introduction of this term to ensure this is more explicit in the revision. A revised version can be seen here at line 86 in our main manuscript:

"In this case, coupled chemical reactions would collapse into a common Arrhenius equation with one master activation energy and integrated frequency factor, which combines each reaction's individual frequency factor into one."

5) Figure 1A: Is it "yolk" instead of "yoke"? "Yoke" is mentioned in the movie as well.

Thank you for pointing this out. We have corrected this mistake throughout the manuscript.

6) Much of the important data are presented in Figures 1C and D, but we found them extremely hard to read, even when they are blown up in one of the supplementary figures. The color scheme and the legend are also amplifying this particular issue. Could the authors plot them with a log scale on the x-axis or as a proportion of total developmental time as was done in the Kuntz and Eisen paper?

Thanks for pointing this out. To address this concern, we have updated figure EV3 A&B and show time in log space to better separate the data. Furthermore, we have increased the opacity of our lines to make them bolder and easier to distinguish from one another. To conserve the developmental time scaling we have chosen to leave our main figure 1 untouched except for the increased opacity on the lines to make them more individually distinct. Changes can be seen in the following plots.

Finally, our previous figures showed time error reported in erroneously 95% CI, which is much larger than typically reported. We have since changed the error bars to show standard deviation, making the individual lines more distinct. These changes can be seen in figures located at lines 649, and 742:

Figure 1: Temperature dependence of development progression in fly and frog embryos. C) Shown here is a schematic depicting how time intervals for stages are determined based on beginning and ending score. Plotted also are all mean time intervals measured at various temperatures ($n = 2$ to 13 per temperature) since $t=0$, 14th syncytial cleavage, in *D. melanogaster* embryos to reach various developmental scores shown in (A). Error bars in time indicate standard deviation among replicates. Error bars in temperature are the standard error given by the manufacturer of 0.5 °C

Figure EV3: Different temperature dependence of developmental progression in frog and fly embryos. A) As Figure 1B but log transformed.

Figure 1: Temperature dependence of development progression in fly and frog embryos. D) As (C) but for *X. laevis*, since T0 (3rd cleavage) at temperatures ranging from 10.3 °C to 33.1 °C (n = 1 to 23 per temperature). Datasets within temperature error were combined and temperature error bars increased to encompass the original error range.

Figure EV3: Different temperature dependence of developmental progression in frog and fly embryos. B) As Figure 1D log transformed.

7) Figure 2C, caption: You're showing the interval, not the endpoint on the x-axis labels, right?

As the reviewer suspects, we are showing the interval (start and end points, for example C-D). We reviewed our material to make sure captions/legends are consistent with the actual figures as they were often not clear. We revised the figure legend of 2C specifically to make this point more clear, adding the following to line 676:

“The x-axis is labeled with the developmental interval, marked by start and endpoint.”

8) Figure 4C: Some graphical way to show the difference in curvatures to make it clear that there is a modest divergence would be helpful.

This is a valid concern that we should have realized earlier. The deviation between 4C and 4D is barely noticeable, additionally the simulated data points further obscure the modest divergence from linearity. To better represent our points we have combined figures 4C & D into one figure with two y-axis to compare the two curves to an overlapping linear fit, as well as removed the superfluous data points.

The original figures and figure legends were as follows:

Figure 4: Complexity and non-idealized behavior of individual enzymes can contribute to non-idealized behavior of developmental processes. B) Shown

is equation (3)’s modeling of a 1000 coupled reaction network with randomly chosen A and E_a . Cyan data points represent the apparent linear temperature range found in our fly data and are fit with a linear regression (solid blue). For comparison also shown is a quadratic fit through the entire temperature range (dashed magenta). Error bars indicate 95% confidence intervals for temperature

based on our experimental setup. **C)** As (B) however optimized A and E_a to maximize the curvature at $T = 295 \text{ }^\circ\text{K}$ (dashed magenta) were passed to equation (3).

The updated figure 4C and figure legend are shown following, as seen at line 702:

Figure 4: Complexity and non-idealized behavior of individual enzymes can contribute to non-idealized behavior of developmental processes. B) Shown is a schematic of a multi-reaction network from Stage 1 to Stage n . Comparison of two reaction networks modeling equation (3) with 1000 coupled reactions, one with randomly selected E_a and A (dashed blue line), the second with E_a and A optimized for maximum curvature at $295 \text{ }^\circ\text{K}$ (dashed orange line). To allow direct comparisons the y-axes were scaled to result in overlapping tangents calculated at $295 \text{ }^\circ\text{K}$ (solid black line).

9) The methods section needs to be significantly overhauled:

9.1) The math is very hard to follow with variables left undefined in the text before they appear in the equations.

Thank you very much for bringing this up. The math in the supplemental material (appendix) has been reformatted and variables are now properly explained in the text when they are first used. Additionally we have made efforts to incorporate math into text more so as the logical flow can be more easily followed. An example of such reformatting from the appendix on line 123 is as follows:

“First, we show that a relaxation time scale formulation can adequately yield the known $\tau=1/k$, where τ is the reaction network’s time constant and k is the rate constant, assuming a simple transition from some stage $A \rightarrow B$. Here, A_0 is the initial amount of A and $B(t)$ is the amount of B at time t , and we define a function $R(t)$, which defines the fraction of the total mass that hasn’t been converted to the final product.”

$$R_{A \rightarrow B}(t) = \frac{A_0 - B(t)}{A_0}$$

If we take the difference between two points on $R(t)$ we get the incremental change:

$$R_{A \rightarrow B}(t) - R(t + dt) = - \frac{\partial R}{\partial t} dt$$

Now we can define the network’s time constant, τ :

$$\tau = \int_0^{\infty} t \left(- \frac{\partial R}{\partial t} \right) dt$$

Integrating we arrive at the following expressions:

$$\begin{aligned} &= tR(t)|_0^{\infty} + \int_0^{\infty} R(t) dt \\ &= \int_0^{\infty} R(t) dt \end{aligned}$$

Before fully solving this integral we must first simplify $R(t)$, where $R(t) = \frac{A_0 - B(t)}{A_0}$. Here we can substitute $A_0 - A_0 e^{-kt}$ for $B(t)$, giving:

$$R(t) = \frac{A_0 - (A_0 - A_0 e^{-kt})}{A_0}$$

And finally:

$$= e^{-kt}$$

9.2) The computational approach is not described in sufficient detail.

We have explained our computational approach in more detail and what these methods are and how they give results. Additionally our code can be found at the following link:

<https://github.com/wuhrlab/ArrheniusAndAnimalDevelopment>. The updated methodology is detailed on line 455 of our main manuscript:

“Using the `fmincon` function and `MultiStart` in matlab a global optimization was performed to maximize the curvature of our sequential linear reaction series equation (3) as a proxy for quantifying nonlinearity. Optimization was done for 2 reactions with constraints on E_a (20-100 kJ (Lepock, 2005)) and on k (1 sec to 3 days). The optimized resulting 2-reaction combination with highest curvature at 295 K was expanded to a 1000 reaction equivalent by expanding the lower E_a reaction to a 999 reaction equivalent with adjusted activation energy. This was possible because multiple Arrhenius reactions can collapse to a single reaction if they share the same E_a . Activation energies were adjusted by subtracting the log of the reaction network size minus 1 all multiplied by RT (the new optimized E_a was calculated as $66.7445806805504 - \log(\text{netSize}-1) \cdot (R \cdot 295.15)$).

Optimized values for E_a and A were then substituted into equation (3) to predict rates at similar temperature points as investigated in our time-lapse experiments for the associated reaction network size as seen in Fig. 4C.

For the random simulated network `rand()` was used to choose random E_a and k within reasonable bounds (Lepock). Reasonable k were determined as the inverse of embryonic states between 1 second and 3 days. `Rand()` results were then fed into equation (3) to predict the overall network as displayed in Fig. 4B”

9.3) Figure S2A: What's PMG?

Thanks for pointing this out. PMG is supposed to mean posterior midgut. We updated the figure legend to better spell this out, seen on line 731:

“Seven additional developmental events (scores) in fly that were later cut. PMG indicates Posterior Midgut.”

9.4) Figure S5A: The labels on the plot are pretty obscure.

Thanks for pointing this out. We have updated the labels on the plots to make them easier to understand. The original S5A is as follows:

Here is the updated Figure S5A, now named Appendix Fig. S3A, with more intelligible names, can be seen on line 53:

Appendix Figure S3: Concavity of developmental intervals and Monte Carlo simulations of Fig. 4C&D show model ability to contribute to nonlinearity.

A) Shown here is a comparison of the frequency of concave downward Arrhenius plots for our biologically scored time-lapse data (over every developmental

interval, fit over the entire temperature range). Additionally shown are frequencies of concave downward Monte Carlo simulations (underlying model is the optimized model from Fig. 4B, propagated errors sourced from biological time-lapse data in time and temperature (entire range)). Shown are means of 100x Monte Carlo simulations for each developmental interval.

9.5) The description of temperature control and monitoring throughout the experiments was also insufficient in our opinion. Why did the authors abandon the microfluidic setup used in the same research groups for previous papers? Further, given that the room temperature was controlled, did the authors also monitor the sample temperature? How is the temperature of acclimated water monitored and maintained?

The microfluidics device is not compatible with frog embryos. As for fly embryos we chose the most stable and consistent method for recording our long developmental videos. For this reason we chose a highly controlled ambient temperature control method to ensure consistent image quality at our high magnification that allowed us to run several experiments simultaneously.

We have updated the description of temperature control and monitoring throughout the experiment in the updated Materials and Methods, as seen at line 429.

“A temperature recorder (Elitech RC-5) was placed near the embryo on the microscope stage to record temperature over the experiment’s duration.”

We have compared our temperature recording method (data logger by the embryo cage) to a record of the solution temperature (using an aquatic thermometer); For some temperatures the results are shown below. Each experiment was performed after allowing the controlled temperature chamber to equilibrate for several hours. There is little to no difference between how the temperature was recorded. For the analysis throughout the paper we used the measured ambient temperature of the stage next to the embryo for both frog and for fly.

Incubator Setting	12.4	21.5	24.5
Water Temp	12.1	21.8	24.4
Air Temp	12.3	21.9	24.4

Legend: Table showing a control test comparing the used ambient temperature recordings against actual temperature of water. Temperatures are shown in $^{\circ}\text{C}$.

Thank you for sending us your revised manuscript. We have now heard back from the three reviewers who were asked to evaluate your study. You will see from the comments below that the reviewers think that while the majority of the concerns have been addressed, several important issues remain. In principle, our editorial policy only allows a single round of major revision. However, we think it is important to address Reviewer #3's concerns with regard to 1) a direct comparison to existing related data and 2) temperature controls, and to discuss Reviewer #1's concerns about the non-physiological temperatures. Therefore, we would ask you to address these points together with all other comments from the three reviewers in an exceptional second round of revision.

On a more editorial level, please do the following.

REFeree REPORTS

Reviewer #1:

The authors have attempted to address my comments and those of the other reviewers. For some comments, they have adequately addressed the issue, such as increasing the number of *Xenopus* replicates to obtain stronger statistics.

Other responses are not sufficient.

1. They tested GAPDH at 2 substrate concentrations to determine if the thermodynamic behavior is concentration dependent. They said they did titrations to ensure substrate was saturating. However nowhere in the manuscript did they present the data. Nor did they do so for the new enzyme they assayed, beta-galactosidase. This data should be added in supplement plus a description as to why it was done and what it means. I will not be the only reader who wonders about the simplicity of the results and they need to make an effort to show they did it;
2. They have STILL not fixed the Chong citation
3. They uploaded the raw imaging data to two separate databases. One, cellimagelibrary.org has the files that can be run but no metadata is available for each movie. Thus I didn't know which temperature or replicate each specimen was being imaged. Deposited data MUST be properly annotated. The data on googledriv.com can only be downloaded to a local computer. The data is huge! I did not download it and so I cannot see whether those were properly annotated. But the data should be accessible and viewable on the cloud and not necessarily on a local computer
4. I did not make a note of the breadth of typos and various errors in the original manuscript but I certainly noticed them. The other reviewers made note of it though, and although the authors say they fixed the errors, if they fixed those, then they made perhaps even more. The manuscript is rife with errors. Sloppy is the word. For example, they sometimes write galactosidase (correct) and sometimes galactidase (not).
5. And by sloppy I am not simply saying they are sloppy with spelling English words. Appendix S4 in the rebuttal they write 0.0028 μ m GAPDH. In the methods they write they used 0.0028mM, which is 2.8 μ M. The whole seems like it was hastily put together without checking and rechecking for errors, which is an essential part of putting a scientific manuscript together. I worry that if a similar cavalier attitude was taken in doing the experiments and their analysis, then it too is filled with errors.

Finally, the authors really do not address the non-physiological temperatures at which the non-linear data reside with respect to the Arrhenius law. For the fly frog and enzyme experiments, the outliers from linearity reside in the extreme temperatures above and below the "sustainable living" range for each animal, ie 30, 37 and 8 C are not sustainable for *Drosophila*. For the enzymes, taken from bacteria or animals that live at a constant 37C, the outliers sit at temperatures of 50, 45, 41C. In the discussion they need to put their claims of regions where development does not fit Arrhenius in this context. Acute temperature stress response utilizing specialized mechanisms may modulate the cell and molecular scale events occurring during development. I know they have a sentence buried amongst some nonsense about rate limiting biochemistry. But effort needs to be made to note the non-physiological range of these behaviors and what that implies. As they stated at one point in the rebuttal, they are more agnostic. That model might be the more correct one than the one they push, and so they should be as agnostic as possible since they have no experiments to support or refute that hypothesis.

In conclusion, the authors could have responded to my comments and those of my colleagues more to heart and really modified the manuscript to thoroughly incorporate the responses into the meat of the manuscript and thereby improve it. The present response is inadequate.

Reviewer #2:

Follow-up Referee Report for: Evaluating the Simple Arrhenius Equation for the Temperature Dependence of Complex Developmental Processes

Crapse et al. have substantially revised the manuscript. Regarding the comments raised by the referees, they have dealt with many of the issues raised and I have more confidence in the results now presented.

I have one specific comment that requires further clarification:

In Appendix Figure S1B, the EA for C-D (I think), suddenly drops to around \sim 55kJ/mol. Yet, this represents a very small time period of the total development of the

Drosophila embryo. I'm still not convinced this observation of decreased EA has any specific physiological relevance. I don't expect a detailed analysis, but it would be good to comment on this more explicitly - currently, the authors only highlight the statistical significance (Fig. 2C). What (if any) biological relevance there is to this observation remains under-developed

Another point is that the references still appear to have errors. Of course, this can be picked up at the proofing stage, but care should be taken that they are all correct.

Reviewer #3:

The authors have taken care of the vast majority of our concerns. Two minor issues remain, however:

1) Why is it so hard to find the Kuntz and Eisen data to perform a direct comparison? A quick search on the Internet gave:

- i) https://figshare.com/articles/dataset/Raw_data_for_Kuntz_and_Eisen_2015_Oxygen_changes_drive_non_uniform_scaling_in_Drosophila_melanogaster_embryo
- ii) <https://github.com/sgkuntz>
- iii) <https://datadryad.org/stash/dataset/doi:10.5061/dryad.s0p50>

Isn't the data there? Did the authors try to contact Mike Eisen?

2) We thank the authors for providing information about their temperature control setup, which was completely missing from the first version of the manuscript. How did they ensure that there is no temperature gradient within the sample holder? Do they have a validation of the temperature the flies feel based on, for example, the timing of the early nuclear divisions?

While we are aware that MSB typically only allows for one round of revisions, we hope that the Editor will agree to let the authors make these minor revisions. We will be happy to look at a next version of the manuscript responding to the remaining issues.

Reviewer #1:

The authors have attempted to address my comments and those of the other reviewers. For some comments, they have adequately addressed the issue, such as increasing the number of *Xenopus* replicates to obtain stronger statistics.

Other responses are not sufficient.

1, They tested GAPDH at 2 substrate concentrations to determine if the thermodynamic behavior is concentration dependent. They said they did titrations to ensure substrate was saturating. However nowhere in the manuscript did they present the data. Nor did they do so for the new enzyme they assayed, beta-galactosidase. This data should be added in supplement plus a description as to why it was done and what it means. I will not be the only reader who wonders about the simplicity of the results and they need to make an effort to show they did it;

We are sorry that in our previous submission the supplementary figure demonstrating that GAPDH and beta-galactosidase were in the saturated regime were not highlighted clear enough (Appendix figures S4 in last submission).

We have modified our text to more clearly reference these additional measurement. The modified text reads as follows:

Old:

“Interestingly, we find that GAPDH shows clearly non-linear behavior in the Arrhenius plot from 10 to 45 °C (Fig 4D, Appendix Figure S4A, Appendix Table S5).”

New, line 284:

“Interestingly, we find that GAPDH shows clearly non-linear behavior in the Arrhenius plot from 10 to 45 °C (Fig 4D). When halving the substrate concentrations used we find similar kinetics, suggesting the enzyme is in the saturated regime (Appendix Figure S6A, Dataset EV6).”

Additionally we have modified our text referring to β -galactosidase:

Old:

“Additionally we have assayed another common enzyme, Beta-galactosidase, monitoring conversion of ortho-Nitrophenyl-Beta-galactoside at 420 nm (Fig. EV5, Appendix Figure S4B, Appendix Table S6, 7), where we find similar results to our GAPDH assay”

New, line 287:

“Additionally, we have assayed another common enzyme, β -galactosidase, monitoring the conversion of ortho-Nitrophenyl- β -galactoside at 420 nm (Fig. EV5, Dataset EV7, 8). Also here, we find that the enzyme shows strong non-linearity in the Arrhenius plot. We performed these experiments with saturating substrate concentrations (Appendix Figure S6B).”

2. They have STILL not fixed the Chong citation

This is embarrassing and we are very sorry for this repeated omission. We originally thought the reviewer was pointing out that there was a software error when linking citations. However, we realized that our citation format was off. We have updated the formatting for the Chong et al citations from :

“Chong, J., Amourda, C., and Saunders, T.E. Temporal development of *Drosophila* embryos is highly robust across a wide temperature range. 11.”

To, line 845:

“Chong J, Amourda C & Saunders TE (2018) Temporal development of *Drosophila* embryos is highly robust across a wide temperature range. *J R Soc Interface* 15: 20180304”

3. They uploaded the raw imaging data to two separate databases. One, cellimagelibrary.org has the files that can be run but no metadata is available for each movie. Thus I didnt know which temperature or replicate each specimen was being imaged. Deposited data MUST be properly annotated. The data on googledriv.com can only be downloaded to a local computer. The data is huge! I did not download it and so I cannot see whether those were properly annotated. But the data should be accessible and viewable on the cloud and not necessarily on a local computer

During the last submission, we were still working with staff from the Cell Image Library to match the metadata with the supplied movies. Because the timing of the update was not under our control, we supplied the movies with matching metadata as a temporary solution via google drive. Since then the Cell Image Library data has been updated. Please refer to our updated data accibility links for our fly and frog movies:

Data Availability:

Fly developmental time-lapses: Cell Image Library server
(<http://cellimagelibrary.org/groups/53322>)

Frog Developmental Time-lapses: Cell Image Library server
(<http://cellimagelibrary.org/groups/54564>)

Modeling and analysis scripts: (<https://github.com/wuhrlab/ArrheniusAndAnimalDevelopment>)

We have updated our frog videos to be clearer which embryo refers to which data in our datasets, to do so we start each video with a second long “key frame” that labels each embryo with a letter to reference them to specific embryos in datasets EV2 and EV4. Additionally we have updated all of our datasets with legends detailing their contents as well as where to find any relevent raw source data. An example for frog from Dataset EV4 is as follows:

“*Xenopus laevis* developmental time data used for analyzing the Arrhenius equation in Frog. Individual sheets contain the temperature of the experiment (the row headed with 'temp', and sheet name) as well as score abbreviations and names (rows 1-2 respectively). Absolute time, time since T0 (accumulated), and time since last score (per-stage) are recorded at each score for each embryo replicate collected at the reported temperature.

Each embryo's data is associated with a movie located here (<http://cellimagelibrary.org/groups/54564>). Movies are labeled in the following format: Xenopus_experimentalTemperatureC_batchNumber. The experimentalTemperatureC correlates to a sheet name. The 'batchNumber' correlates to the run, where if multiple experiments were conducted at the same temperature then sheets are named as 'temperatureRNumber', where RNumber matches batchNumber. Specific embryo identifiers are located in column 'A' and correlate to the beginning frame of each relevant video. Each movie is given a CIL# which is recorded under "temp" on each sheet."

As for our fly datasets we have updated them similarly to the following, an example from Dataset EV3:

"*Drosophila melanogaster* developmental time data used for analyzing the Arrhenius equation in Fly. Individual sheets contain the temperature of the experiment (the row headed with 'temp', and sheet name) as well as score abbreviations and names (rows 1-2 respectively). Absolute time, time since T0 (accumulated), and time since last score (per-stage) are recorded at each score for each embryo replicate collected at the reported temperature

Each embryo's data is associated with a movie located here (<http://cellimagelibrary.org/groups/53322>). Movies are labeled in the following format: Drosophila_experimentalTemperatureC_embryoID. The experimentalTemperatureC correlates to a sheet name. The 'embryoID' then correlates to the letters in column 'A' under 'Embryo Name'. Each movie is given a CIL# which is recorded under each associated embryo name in column A."

Unfortunately, the Cell Imaging Library website does not offer high-quality previews for movies. To view our data in the original quality one will need to download it.

4. I did not make a note of the breadth of typos and various errors in the original manuscript but I certainly noticed them. The other reviewers made note of it though, and although the authors say they fixed the errors, if they fixed those, then they made perhaps even more. The manuscript is rife with errors. Sloppy is the word. For example, they sometimes write galactosidase (correct) and sometimes galactidase (not).

Thanks for pointing this out. Based on the reviewers' comments, we asked a professional science writer to edit the manuscript before this resubmission.

5. And by sloppy I am not simply saying they are sloppy with spelling English words. Appendix S4 in the rebuttal they write 0.0028 μ m GAPDH. In the methods they write they used 0.0028mM, which is 2.8 μ M. The whole seems like it was hastily put together without checking and rechecking for errors, which is an essential part of putting a scientific manuscript together. I

worry that if a similar cavalier attitude was taken in doing the experiments and their analysis, then it too is filled with errors.

We apologize for the many typos. The reviewer is correct that we were indeed in time-trouble to meet the journal's resubmission deadline and did not spend as much time as we should to polish the manuscript. We have hopefully done a better job this time and additionally asked a professional science editor to rectify our mistakes. We are particularly sorry for accidentally using mM where we should have used μM .

Correction can be seen on line 597 of the main text:

"0.0024 μM "

Finally, the authors really do not address the non-physiological temperatures at which the non-linear data reside with respect to the Arrhenius law. For the fly frog and enzyme experiments, the outliers from linearity reside in the extreme temperatures above and below the "sustainable living" range for each animal, ie 30, 37 and 8 C are not sustainable for *Drosophila*. For the enzymes, taken from bacteria or animals that live at a constant 37C, the outliers sit at temperatures of 50, 45, 41C. In the discussion they need to put their claims of regions where development does not fit Arrhenius in this context. Acute temperature stress response utilizing specialized mechanisms may modulate the cell and molecular scale events occurring during development. I know they have a sentence buried amongst some nonsense about rate limiting biochemistry. But effort needs to be made to note the non-physiological range of these behaviors and what that implies. As they stated at one point in the rebuttal, they are more agnostic. That model might be the more correct one than the one they push, and so they should be as agnostic as possible since they have no experiments to support or refute that hypothesis.

We apologize for the confusing terminology used in our paper and less than optimal clarity of data labeling in the Arrhenius plot. In the revised version we have indicated the core temperature range (14.3 °C to 27 °C in *drosophila*, and 12.2 °C to 25.7 °C in frog) as the approximate linear range we use for a linear fit to deduce the apparent activation energies. In the previous submission, all data within this regime were labeled in blue in our Arrhenius plots and data-points outside this regime in red. However, the range at which embryos survive to the last stage scored in our assays ("First Breath" in fly and "Late Neurulation" in frog) is different. This viable temperature range is wider than this core temperature range (14.3 °C to 30.1 °C in fly and 12.2 °C to 28.5 °C in frog) and shows clear non-linearity. We added a separate Appendix Figure redoing the analysis for non-linearity for only this viable temperature range (Appendix Figure S4). To make this clearer to the readers we updated all Arrhenius plots in the paper so that blue data-points indicate the viable temperature range, while the red data-points indicate temperatures in which the embryos are viable for some stages of development but not at others. We also want to point out that the embryos ability to withstand very low or high temperatures even for only parts of developmental progression is very likely to be physiologically relevant for exothermic species like frog and flies as this could increase their chances to survive temporary temperature fluctuations.

Old:

“Both the frog and fly data exhibit wide core temperature regions that we approximate with a linear fit, between 14.3 and 27 °C in flies and 12.2 and 25.7 °C in frogs (Fig. 2A, B, Fig. EV3). However, for each organism clear deviations are observed as temperatures near the limits of the viable range and outside the core temperatures (Fig. EV3C, D).”

New, Line 140:

“Both the frog and fly data exhibit wide core temperature regions that we approximate with a linear fit, between 14.3 and 27 °C in flies and 12.2 and 25.7 °C in frogs (Fig. 2A, B, Fig. EV2). However, for each organism we observe clear deviations from linearity, particularly outside of these temperature ranges (Fig. EV2) “

Additionally, we have added Appendix Figure S4 to specifically show the non-linearity of the ‘viable temperature range’. We still observe strong quadratic behavior, which shows that this behavior is not limited to ‘non-physiological’ or ‘non-viable’ temperatures, but rather is a characteristic throughout. We understand that confusion may arise due to our claims of linearity earlier in the paper. However, these claims were made because the data appeared strikingly linear in the Arrhenius plot, especially compared to the most extreme, non-viable temperatures.

We re-wrote the associated paragraph in the main text on line 212 to better explain our position:

“These findings raise the question if the temperature region is also non-linear for the temperature range over which the embryos can develop to the last scored developmental event (14.3 to 30.1 °C in fly and 12.2 to 28.5 °C in frog). We performed BIC analysis for all developmental intervals in fly embryos and find this “viable regime” is clearly quadratic over most intervals (Appendix Figure S4B, C). Although less conclusive, we find similar results when reanalyzing our frog data (Appendix Figure S4D, E). We always observe deviation to be downward concave, i.e. the rates at very low and very high temperatures are lower than predicted by the Arrhenius equation.

Thus, while the Arrhenius equation is a good approximation for the temperature dependence of early fly and frog development, at temperature extremes, we see clear deviation. This observation supports our initial intuitions that Arrhenius cannot perfectly describe a complex system; although why it deviates and how it is still a fairly decent approximation remains to be answered..”

The new Appendix Figure S4 is as follows:

Appendix Figure S4: ANCOVA to compare activation energies and BIC to compare quadratic versus linear fit over the viable temperature range. A) p-values for fly developmental stages using ANCOVA calculated to determine the probability of observing difference between activation energies for every combination of fly developmental intervals shown in Fig. EV2C. Blue marks p-values above 5E-2, purple marks $\leq 5E-2$, pink marks $\leq 1E-2$, and red marks $\leq 1E-3$. **B)** Linear (black blue) and quadratic (dashed red) fits calculated for an example fly developmental intervals from 14th Cleavage to Beginning of Germband Retraction over the viable temperature range where embryos survive until First Breath (blue data, excluding red data). BIC was used to test model preference for a quadratic fit, reported in the top right as the log ratio likelihood for quadratic over linear for this temperature range. **C)** Shown is a heatmap reporting the natural log ratio likelihood of quadratic over linear fit preference for all developmental intervals in fly development over the viable temperature range. Intervals are marked with their beginning event on the X-axis, and their ending event on the Y-axis. Red represents a preference for quadratic while blue represents a preference for a linear fit. **D)** As (A) but for calculating frog p-values between slopes of developmental intervals shown in Fig. EV2D. **E)** As (B) but for frog 3rd to 10th cleavage over the viable temperature range where embryos survive until End of Neurulation. **F)** As (C) but for all possible frog intervals.

Lastly, we have extended the discussion based on the reviewer's comment and thank him for the sentence: "Acute temperature stress response utilizing specialized mechanisms may modulate the cell and molecular scale events occurring during development.", which we have adapted to our manuscript.

Old:

"However, there remain several additional possibilities contributing to this highly complex behavior. For example embryos might activate entirely different pathways at certain temperatures e.g. via cold or heat stress. Additionally, single

multi-step processes (e.g. transcription) might exhibit different rate-limiting steps at different temperatures due to the different underlying activation energies (Roe et al., 1985).”

New, Line 334:

“However, several other factors may also contribute to this behavior. Although we have observed nonlinearity over temperature ranges where morphology is normal and viability is high, it is possible that additional processes come into play at extreme temperatures. Acute temperature stress responses utilizing specialized mechanisms may modulate the cell and molecular scale events occurring during development. Embryos might activate entirely different pathways at more extreme, near non-viable, temperatures e.g. via cold or heat stress.”

In conclusion, the authors could have responded to my comments and those of my colleagues more to heart and really modified the manuscript to thoroughly incorporate the responses into the meat of the manuscript and thereby improve it. The present response is inadequate.

We are very sorry that we seem to have left this impression. We greatly appreciate the reviewers' constructive criticism and have tried our best to address their comments to improve the manuscript. We believe a track change comparison between of manuscripts from the different submissions reveals that we have drastically updated the manuscript and added substantial amounts of additional data and analysis. We believe both rounds of reviews have improved the manuscript and we hope that the additional improvements and clarifications put forward in this round are able to address the remaining concerns.

Reviewer #2:

Follow-up Referee Report for: Evaluating the Simple Arrhenius Equation for the Temperature Dependence of Complex Developmental Processes

Crapse et al. have substantially revised the manuscript. Regarding the comments raised by the referees, they have dealt with many of the issues raised and I have more confidence in the results now presented.

I have one specific comment that requires further clarification:

In Appendix Figure S1B, the EA for C-D (I think), suddenly drops to around ~55kJ/mol. Yet, this represents a very small time period of the total development of the *Drosophila* embryo. I'm still not convinced this observation of decreased EA has any specific physiological relevance. I don't expect a detailed analysis, but it would be good to comment on this more explicitly - currently, the authors only highlight the statistical significance (Fig. 2C). What (if any) biological relevance there is to this observation remains under-developed

Thanks for pointing this out. We would like to point out the analysis of our frog data reveals several stages with much longer time-intervals to be statistically significant different (Appendix Figure S3 & S4). Further, when we reanalyze the data from Kuntz & Eisen (Fig. EV3) activation energies of larger regimes also appear to be significantly different. The physiological relevance is probably minimal if all developmental events with different activation energies would indeed be entirely sequential. For technical reasons, in this study we can only score sequential events. However, we are very puzzled by how embryos could deal with differential scaling with temperatures for processes that run in parallel. Interestingly, it has been previously been demonstrated that even different stages of the same cell cycle show different activation energies (PMID: 33278404). These events are temporally coupled closely making it likely that they occur at least somewhat in parallel. To discuss this, we have updated the discussion section of the manuscript as follows:

Line 341:

“One major question this study raises is how complex embryonic development can result in a canonically developed embryo if the different reactions required for faithful development proceed at different relative speeds at different temperatures. In our assays, we are only able to follow temporally sequential reactions, and one can argue that increasing or decreasing time spent at a particular event should not influence the success of development. However, development must be much more complex and hundreds or thousands of reactions and processes must occur in parallel, e.g. in different cell types developing at the same stage. Therefore, how can frog and fly embryos be viable over a ~15° C temperature range wherein different developmental intervals’ varying temperature sensitivity could possibly throw development out of balance? We envision two major possible developmental strategies to overcome this problem. Either all rate-limiting steps occurring in parallel at a given embryonic stage have evolved similar activation energies, or the embryos have developed checkpoints that assure a resynchronization of converging developmental processes over wide temperature ranges.”

Furthermore we have revised our original Appendix Figure S1 (now S3) for clarity and flow as the developmental intervals of interest was not legible (the significant score in D-E, not C-D). Additionally we have made the length of the interval as a portion of development more clear. The old and revised figures can be found as follows:

Old:

Appendix Figure S1: Distribution of bootstrapped apparent activation energies for various developmental periods in fly and frog. A) Histograms of apparent activation energies in Fly calculated from 5000x bootstrapped fits on data used to generate Fig. 2C. Displayed also is the median (dashed red line) as well as 68% confidence (dashed black line). **B)** As Fig. 2C, but the x-axis has been scaled by mean developmental timings of each stage at 21.1 °C. **C)** As (A) but bootstrapping was performed on Frog data used to generate Fig. 2D. **D)** As Fig. 2D, but the x-axis has been scaled by mean developmental timings of each stage at 22.2 °C.

New:

Appendix Figure S3: Apparent activation energies spaced by developmental time and distribution of bootstrapped apparent activation energies.

A) As Fig. 2C, but the x-axis has been scaled by mean developmental timings of each stage at 21.1 °C. Grey vertical lines mark the borders of the developmental intervals as a portion of development from Z - K. **B)** As Fig. 2D, but the x-axis has been scaled by mean developmental timings of each stage at 22.2 °C. Grey vertical lines mark the borders of the developmental intervals as a portion of development from A - L. **C)** Histograms of bootstrapped apparent activation energies in fly calculated from 5000x bootstrapped fits on data used to generate Fig. 2C. Displayed is the median (dashed red line) as well as 68% confidence intervals (dashed black line). **D)** As (B) but bootstrapping was performed on frog data used to generate Fig. 2D.

Another point is that the references still appear to have errors. Of course, this can be picked up at the proofing stage, but care should be taken that they are all correct.

We are very sorry for this. We noted that some of our references were incorrectly stored in our reference software due to manually adding references. For this submission, we have updated all reference information via Zoteros' "add item by identifier", using each paper's DOI or PMID (for Chong et al: <http://dx.doi.org/10.1098/rsif.2018.0304>). Shown below is an example of the updated references.

Old:

"Chong, J., Amourda, C., and Saunders, T.E. Temporal development of *Drosophila* embryos is highly robust across a wide temperature range. 11."

New, line 845:

"Chong J, Amourda C & Saunders TE (2018) Temporal development of *Drosophila* embryos is highly robust across a wide temperature range. *J R Soc Interface* 15: 20180304"

Reviewer #3:

The authors have taken care of the vast majority of our concerns. Two minor issues remain, however:

1) Why is it so hard to find the Kuntz and Eisen data to perform a direct comparison? A quick search on the Internet gave:

i)

https://figshare.com/articles/dataset/Raw_data_for_Kuntz_and_Eisen_2015_Oxygen_changes_drive_non_uniform_scaling_in_Drosophila_melanogaster_embryogenesis_/1582639

ii) <https://github.com/sgkuntz>

iii) <https://datadryad.org/stash/dataset/doi:10.5061/dryad.s0p50>

Isn't the data there? Did the authors try to contact Mike Eisen?

Thanks for providing these links. We have investigated those before but those do not provide access to the data of interest: the first link refers to data from Kuntz & Eisen's paper investigating oxygen's effects on development (PMID: [26673611](https://pubmed.ncbi.nlm.nih.gov/26673611/)), rather than the paper we reference on temperature's effects on Development (PMID: [24762628](https://pubmed.ncbi.nlm.nih.gov/24762628/)). The second link points indeed to the *drosophila melanogaster* videos one would need to make a direct comparison. But unfortunately the link contains only example videos for each temperature but not the full dataset. The third link holds similar data as the second link, where although more expansive containing a few more example fly videos, nowhere near the whole dataset.

We therefore requested the data from the authors. They kindly provided tabulated data for his their developmental scorings for all investigated temperatures, which we reformatted and provide Dataset EV5. Using this data, we investigated if 1) their data supports different activation energies for different developmental intervals. 2) If the data supports a quadratic over a linear fit in the Arrhenius plot. These results are shown in figure EV3, found below. While the developmental scores used by Kuntz & Eisen were different from our own, the reanalysis further supports our main conclusions. Using the Kuntz & Eisen data still show significantly different activation energies and a quadratic fit in the Arrhenius plot is statistically preferred over a linear fit:

Figure EV3. Re-analysis of temperature dependence data in *D. melanogaster* development from Kuntz & Eisen 2014.

A) Scores Kuntz & Eisen used and abbreviation for the remainder of this figure.

B) Arrhenius plots for adjacent stage durations of Kuntz & Eisen's data. We fit data for temperatures below 28.75° Celcius (dashed red line i.e. core temperature range) via linear regression (solid blue line). Shown is the apparent activation energy plus/minus the confidence interval. Additionally, we fit the entire temperature range with quadratic (dashed blue line) and linear fits. BIC was also calculated and is shown here as the natural log ratio likelihood for quadratic over linear fit and displayed in black.

C) Apparent activation energies over the core temperature regime are shown. Error bars show the 68% confidence intervals. Black braces point out example developmental intervals that have significantly different apparent activation energies. *** p-value < 0.01, **** p-value < 0.001.

D) Shown are p-values between all developmental intervals, stage 1 and 2. Blue marks p-values above 5E-2, purple marks $\leq 5E-2$, pink marks $\leq 1E-2$, and red marks $\leq 1E-3$.

E) Shown are natural log ratio of likelihoods for quadratic over linear fits for all possible developmental intervals, marked by their starting and ending scores, using Kuntz & Eisen's data over all temperatures. Blue signifies a preference for linearity; red signifies a preference for quadratic behavior.

We have modified the manuscript to cite the added analysis

lines 152

“In this respect our results differ from the uniform scaling proposed for fly development in a previous study (Kuntz & Eisen, 2014). However, when we re-analyzed the data that the authors kindly provided, we find that apparent activation energies between developmental intervals vary significantly (p -value = 1×10^{-3}) (Fig. EV3A-D, Dataset EV5).”

and 317:

“One striking finding of our study is that different developmental processes within the same embryo clearly scale differently with varying temperature i.e. the apparent activation energies for different developmental intervals can vary significantly. We reaffirmed this observation upon reanalyzing Kuntz & Eisen's 2014 data (Figure EV3). Different temperature scaling has also previously been observed in component processes in presumed simpler processes such as cell cycle progression during the cleavage division in fly embryos (Falahati *et al*, 2020).”

Different E_a s imply different developmental scaling, rather than the uniform developmental scaling posited by Kuntz & Eisen. To resolve this discrepancy we revisited figure 3C in Kuntz & Eisen. At first glance the figure does suggest approximately even scaling across all temperatures for different stages of development. However when we replotted the data in a similar fashion fitting the data with linear fits, as well as slightly stretching the figure (so as to better see individual developmental events) we begin to see a few events which clearly do not share the same vertical lines as the 0 and 1 normalized reference events. The remade figure can be seen as follows, compared against the original Eisen figure 3C:

Figure 3. Developmental time of *D. melanogaster* varies with temperature. [...] (B) The time individual animals reached the various time-points are shown, with each event being a different color. Time 0 is defined as the end of cellularization, when the membrane invagination reaches the yolk. Between 17.5°C and 27.5°C the total time of embryogenesis, t_{dev} measured as the mean time between cellularization and trachea fill, has a logarithmic relationship to temperature described by $t_{dev} \sim 4.02e^{37/31} = T$ where T is temperature in °C ($R^2 \sim 0.963$). (C) The developmental rate in *D. melanogaster* changes uniformly with temperature, not preferentially affecting any stage. Timing here is normalized between the end of cellularization and the filling of the trachea. From doi:10.1371/journal.pgen.1004293.g003

Replotted Kuntz & Eisen figure 3C. Events are colored to match the color coding used in Kuntz & Eisen's figure 3C. Overlaying the data are linear fits. The solid line fits all temperatures (used in our BIC calculations above), the dashed line includes all data at and below 27.5 °C (used for our E_a calculations above. Plotting the data in this manner

intuitively suggests that different developmental periods of fly development scale differently with temperature. This intuition is supported by the F-test analysis in the Arrhenius plots. We find some significant different slopes i.e. activation energies (Fig. EV3 B, C, D).

2) We thank the authors for providing information about their temperature control setup, which was completely missing from the first version of the manuscript. How did they ensure that there is no temperature gradient within the sample holder? Do they have a validation of the temperature the flies feel based on, for example, the timing of the early nuclear divisions?

Thanks for pointing out that we needed to provide more detail on our temperature measurements and control. To this end we added additional information in our Materials and Methods for both fly and frog experiments. For fly, found on line 377 in the main text:

“To record and validate temperatures for the fly embryo data collections, temperatures were taken next to each a microscope’s sample holders (~1 inch from the embryo) using either an Elitech RC-5 (standard error +/- 0.5 °C), Dickson TH300 (standard error, +/- 1.0 °C), or Fluke 54 II B (standard error, +/- 0.3 °C) thermometer. We worked with two microscopes in the room. When comparing the temperatures between microscopes they never differed more than a degree suggesting the temperature in the room was very homogenous.”

When comparing the temperatures at the two microscopes they never differed more than a degree at a minimum separation of 3 feet. Making the maximum possible gradient between thermometer and embryo likely ~1/36th of a degree between the embryo and adjacent temperature recorder.

For our frog acquisitions we added a detailed description of our image capturing and temperature control to the Material and Methods, on line 499 of our main text.

“To validate the temperature experienced by our frog embryos, we used an aquatic thermometer (QTI, DTU6024C-004-C, tolerance provided by the manufacturer +/- 0.1 °C) that measured the temperature of the 0.1 MMR the embryos were raised. Additionally, we recorded the temperature of the surrounding air in the aforementioned temperature controlled chamber with an Elitech RC-5 temperature recorder (+/- 0.5 °C). We observed that these readings agreed with each other within the standard errors of the thermometers. Each experiment was performed after allowing the controlled temperature chamber to equilibrate for several hours. For the analysis throughout the paper we used the measured ambient temperature at the microscope stage, directly adjacent to the frog embryos..”

Representative temperature recording comparisons for evaluating the frog set-up are shown in the table below.

Incubator Setting	12.4	21.5	24.5
Water Temp	12.1	21.8	24.4
Air Temp	12.3	21.9	24.4

Legend: Table showing a control test comparing the used ambient temperature recordings against actual temperature of water for our frog embryo imaging setup. Temperatures are shown in °C.

While we are aware that MSB typically only allows for one round of revisions, we hope that the Editor will agree to let the authors make these minor revisions. We will be happy to look at a next version of the manuscript responding to the remaining issues.

Thank you for sending us your revised manuscript . We have now heard back from the two reviewers who agreed to evaluate your study. As you will see below, the reviewers are satisfied with the modifications made and think that the study is now suitable for publication.

Before we can formally accept your manuscript, we would ask you to address the following issues.

REFEREE REPORTS

Reviewer #1:

The authors have made all of the changes requested. The manuscript is very much improved. It is much clearer and reads very well. Addition of the Eisen data substantiates their findings. Congratulations on an interesting paper!

Reviewer #3:

The authors have taken care of all our concerns.

The authors have made all requested editorial changes.

Thank you again for sending us your revised manuscript. We are now satisfied with the modifications made and I am pleased to inform you that your paper has been accepted for publication.

Corresponding Author Name: Eric Wieschaus, Martin Wühr

Manuscript Number: MSB-20-9895